# Accuracy and social motivations shape judgements of (mis)information

Steve Rathje [1] ✉, Jon Roozenbeek [1], Jay J. Van Bavel [2] & Sander van der Linden [1] ✉

The extent to which belief in (mis)information reflects lack of knowledge versus a lack of motivation to be accurate is unclear. Here, across four experiments ($n$ = 3,364), we motivated US participants to be accurate by providing financial incentives for correct responses about the veracity of true and false political news headlines. Financial incentives improved accuracy and reduced partisan bias in judgements of headlines by about 30%, primarily by increasing the perceived accuracy of true news from the opposing party ($d$ = 0.47). Incentivizing people to identify news that would be liked by their political allies, however, decreased accuracy. Replicating prior work, conservatives were less accurate at discerning true from false headlines than liberals, yet incentives closed the gap in accuracy between conservatives and liberals by 52%. A non-financial accuracy motivation intervention was also effective, suggesting that motivation-based interventions are scalable. Altogether, these results suggest that a substantial portion of people's judgements of the accuracy of news reflects motivational factors.

Misinformation—which can refer to fabricated news stories, false rumours, conspiracy theories or disinformation—can have serious negative effects on society and democracy[1,2]. Misinformation exposure can reduce support for climate change[3,4] or lead to vaccine hesitancy[5–7], and the mere repetition of misinformation can increase belief in it[8,9]. There has thus been a growing interest in understanding the psychology of belief in misinformation and how to mitigate its spread[1,2,10–12].

There are large partisan differences in how people judge information to be true or false. People are much more likely to believe news with politically congruent content[13–16] or news that comes from politically congruent sources[17,18]. However, there are multiple possible reasons that can explain why this partisan divide exists. One possible explanation is that people tend to engage in politically motivated cognition[19,20]: although people are often motivated to be accurate, they also have social goals (for example, group belonging, status and so on) for holding certain beliefs that can interfere with accuracy goals[13]. Another potential explanation is that partisans have different pre-existing knowledge, or different prior beliefs, as a result of exposure to different partisan news outlets and social media feeds[10].

Yet, it is challenging to differentiate between these explanations unless accuracy or social motivations are experimentally manipulated[21–24]. If belief in misinformation in part reflects motivational factors, experimentally manipulating people's accuracy or social motivations should shift people's judgements of misinformation[21,23–25]. However, if belief in misinformation simply reflects different prior beliefs or exposure to different information sources, these experimental manipulations should not change people's judgements of misinformation.

Several studies have also found that US conservatives or Republicans tend to believe in and share more misinformation than US liberals or Democrats[26–32]. One interpretation behind this asymmetry is that US conservatives are exposed to more low-quality information and thus have less accurate political knowledge, perhaps due to US conservative politicians and news media sources sharing less accurate information[33,34]. Another interpretation again focuses on motivation, suggesting that US conservatives may, in some contexts, have greater motivations to believe ideologically or identity-consistent claims that could interfere with their motivation to be accurate[31,35–37]. But, again, it is

[1]Department of Psychology, University of Cambridge, Cambridge, UK. [2]Department of Psychology and Center for Neural Science, New York University, New York, NY, USA. ✉e-mail: srathje@alumni.stanford.edu; sander.vanderlinden@psychol.cam.ac.uk

difficult to disentangle these two explanations without experimentally manipulating motivations.

In this Article, we examine the causal role of accuracy motives in shaping judgements of true and false political news via the provision of financial incentives for correctly identifying accurate headlines. Prior research using financial incentives for accuracy has yielded mixed results. For example, previous studies have found that financial incentives to be accurate can reduce partisan bias about politicized issues[38,39] and news headlines[40], and improve accuracy about scientific information[41]. However, another study found that incentives for accuracy can backfire, increasing belief in false news stories[14]. Incentives also do not eliminate people's tendency to view familiar statements[42,43] or positions for which they advocate[44] as more accurate, raising questions as to whether incentives can override the heuristics people use to judge truth[45]. These conflicting results motivate the need for a systematic investigation of when and for whom various motivations influence belief in news.

We also examine whether social identity-based motivations to identify posts that will be liked by one's political in-group interfere with accuracy motivations. On social media, content that appeals to social-identity motivations, such as expressions of out-group derogation, tends to receive high engagement online[46–48]. False news stories may be good at fulfilling these identity-based motivations, as false content is often negative about out-group members[26,49]. The incentive structure of the social media environment draws attention to social motivations (for example, receiving social approval in the form of likes and shares), which may lead people to give less weight to accuracy motivations online[50,51]. As such, it is important to understand how these social motivations might compete with accuracy motivations[13].

Finally, we compare the effect of accuracy motivations with the effects of other factors that are regularly invoked to explain the belief and dissemination of misinformation, such as analytic thinking[52] political knowledge[53], media literacy skills[54] and affective polarization[49]. By including these variables in the same study, we are able to compare different theoretical models of (mis)information belief and sharing[2,11].

## Overview

Across four pre-registered experiments, including a replication with a nationally representative US sample, we test whether incentives to be accurate improve people's ability to discern between true and false news and reduce partisan bias (experiment 1). Additionally, we test whether increasing partisan identity motivations by paying people to correctly identify posts that they think will be liked by their political in-group (mirroring the incentives of social media) reduces accuracy (experiment 2). Further, we examine whether the effects of incentives are attenuated when partisan source cues are removed from posts (experiment 3). Then, to test the generalizability of these results and help rule out alternate explanations, we test whether increasing accuracy motivations through a non-financial accuracy motivation intervention also improves accuracy (experiment 4). Finally, in an integrative data analysis (IDA), we examine whether motivation helps explain the gap in accuracy between conservatives and liberals, and compare the effects of motivation with the effects of other variables known to predict misinformation susceptibility.

## Results

### Experiment 1: incentives improve accuracy and reduce bias

In experiment 1, we recruited a politically balanced sample of 462 US adults via the survey platform Prolific Academic[55]. Participants were shown 16 pre-tested news headlines with an accompanying picture and source (similar to how a news article preview would show up on someone's Facebook feed). In a pre-test, eight headlines (four false and four true) were rated as more accurate by Democrats than Republicans, and eight headlines (four false and four true) were rated as more accurate by Republicans than Democrats[56]. An example of a Democrat-leaning

true headline was 'Facebook removes Trump ads with symbols once used by Nazis' from apnews.com, and an example of a Democrat-leaning false news headline was 'White House Chef Quits because Trump Has Only Eaten Fast Food For 6 Months' from halfwaypost.com. After seeing each headline, participants were asked 'To the best of your knowledge, is the claim in the above headline accurate?' and were then asked 'If you were to see the above article on social media, how likely would you be to share it?' For more details, see Methods.

Half of the participants were randomly assigned to the 'accuracy incentives' condition. In this condition, participants were told they would receive a small bonus payment of up to one US dollar based on how many correct answers they could provide regarding the accuracy of the articles. The other half of participants were assigned to a 'control' condition in which they were asked the same questions about accuracy and sharing without any incentive to be accurate.

We first examined whether accuracy incentives improved truth discernment, or the number of true headlines participants rated as true minus the number of false headlines participants rated as true[15]. As predicted, participants in the accuracy incentives condition (mean ($M$) = 3.01, 95% confidence interval (CI) 2.68–3.34) were better at discerning truth than those in the control condition ($M$ = 2.43, 95% CI 2.12–2.73), $t(457.64) = 2.58$, $P = 0.010$, $d = 0.24$. In other words, participants answered 11.01 (out of 16) questions correctly in the accuracy incentives condition, as opposed to 10.43 (out of 16) questions in the control condition.

We next examined whether incentives decreased partisan bias, or the number of politically congruent headlines participants rated as true minus the number of politically incongruent headlines participants rated as true. This measurement of partisan bias follows recommendations from prior work[15,57], yet we discuss alternative ways to measure partisan bias and debates about the term 'partisan bias'[58] in Supplementary Appendix 1. We also re-analysed our data using an alternate measure of partisan bias in Supplementary Appendix 1 and found no changes to our main conclusions.

As predicted, partisan bias, or one's belief in politically congruent over politically incongruent claims, was 31% smaller in the accuracy incentives condition ($M$ = 1.31, 95% CI 1.04–1.58) as compared with the control condition ($M$ = 1.91, 95% CI 1.62–2.19), $t(495.8) = 3.01$, $P = 0.001$, $d = 0.28$. Results from all four studies are plotted visually in Fig. 1.

Additional analysis (for extended results, see Supplementary Appendix 1) found that the accuracy incentives condition increased the percentage of politically incongruent true headlines rated as true ($M$ = 51.53%, 95% CI 47.36–55.70) as compared with the control condition ($M$ = 38.25%, 95% CI 34.41–42.08), $P < 0.001$, $d = 0.43$. Incentives did not statistically significantly impact judgements of politically congruent true news, politically incongruent false news or politically congruent false news when controlling for multiple comparisons with Tukey post-hoc tests ($ps > 0.444$). Thus, the effects of incentives were mainly driven by an increased belief in true news from the opposing party.

Finally, we examined whether the incentives influenced sharing discernment, or the number of true headlines shared minus the number of false headlines people intended to share. Interestingly, even though sharing higher-quality articles was not explicitly incentivized, sharing discernment was slightly higher in the accuracy incentive condition ($M$ = 0.38, 95% CI 0.28–0.48) as compared with the control condition ($M$ = 0.22, 95% CI 0.15–0.30), $t(424.8) = 2.49$, $P = 0.037$, $d = 0.23$.

### Experiment 2: social motivations

In experiment 2, we aimed to replicate and extend on the results of experiment 1 by examining whether social or partisan motivations to correctly identify articles that would be liked by one's political in-group might interfere with accuracy motives. We recruited another politically balanced sample of 998 US adults (Methods). In addition to the accuracy incentives and control condition, we added a 'partisan sharing' condition, whereby participants were given a financial incentive

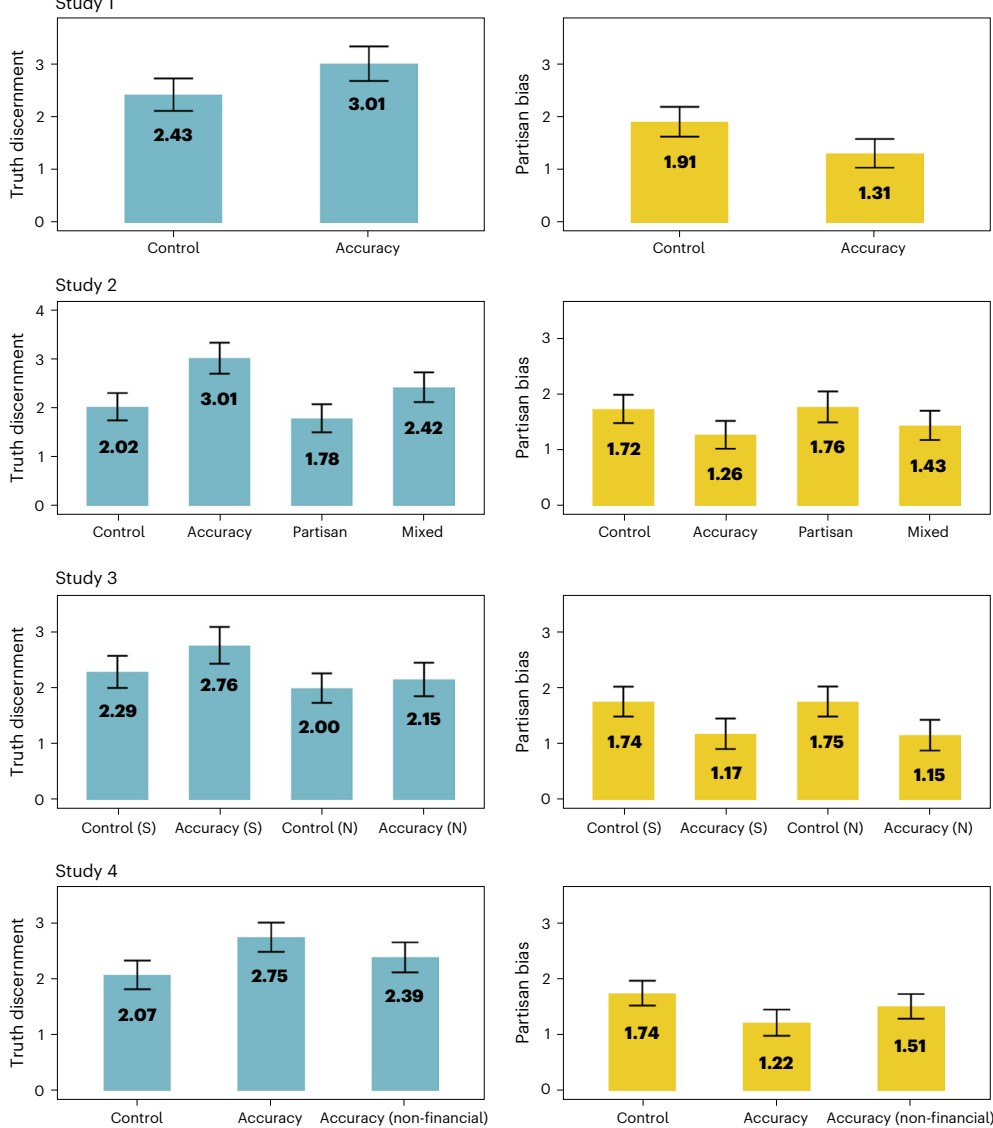

**Fig. 1 | Accuracy incentives improved truth discernment and reduced partisan bias across four experiments.** In study 1 ($n$ = 462), accuracy incentives improved truth discernment and decreased partisan bias in accuracy judgements. Study 2 ($n$ = 998) replicated these findings, but also found that incentives to determine which articles would be liked by their political in-group if shared on social media decreased truth discernment, even when paired with the accuracy incentive (the 'mixed' condition). Study 3 ($n$ = 921) further replicated these findings and examined how effect sizes differed with and without source cues (S, source; N, no source). Study 4 ($n$ = 983) also replicated these findings

and found that that a scalable, non-financial accuracy motivation intervention was also able to increase belief in politically incongruent true news with a smaller effect size. Means for each condition are shown in the figure, and error bars represent 95% confidence intervals. The $y$ axis of graphs on the left represent truth discernment (or the number of true claims rated as true minus the number of false claims rated as false). The $y$ axis of graphs on the right represent partisan bias (or the number of politically congruent claims rated as true minus the number of politically incongruent claims rated as true).

to correctly identify articles that would appeal to members of their own political party. This condition was meant to mirror the incentive structure of social media whereby people try to share content that will be liked by their friends and followers. Specifically, participants were told that they would receive a bonus payment of up to one dollar based on how accurately they identified articles that would be liked by members of their political party if they shared them on social media. Immediately after answering this question, participants were asked about the accuracy of the article and how likely they would be to share it. To examine how partisan identity goals might interfere with accuracy goals, we added a final condition, called the mixed motivation condition, in which participants received a financial incentive of up to one dollar to identify articles that would be liked by one's in-group,

followed by an additional financial incentive to accurately identify true and false articles.

We first examined how these motivations influenced truth discernment. Replicating the results of experiment 1, there was a significant main effect of the accuracy incentives condition on truth discernment, $F(1, 994) = 29.14$, $P < 0.001$, $\eta^2_G = 0.03$, a significant main effect of the partisan sharing manipulation on truth discernment, $F(1, 994) = 7.53$, $P = 0.006$, $\eta^2_G = 0.01$, but no significant interaction between the accuracy and the partisan sharing manipulation ($P = 0.237$). Tukey honestly significant difference (HSD) post-hoc tests indicated that truth discernment was higher in the accuracy incentives condition ($M = 3.01$, 95% CI 2.69–3.32) compared with the control condition ($M = 2.02$, 95% CI 1.74–3.30), $P < 0.001$, $d = 0.41$. Truth discernment was also higher in

the accuracy incentives condition compared with the partisan sharing condition ($M = 1.78$, 95% CI 1.49–2.07), $P < 0.001$, $d = 0.50$, and the mixed condition ($M = 2.42$, 95% CI 2.11–2.71), $P = 0.029$, $d = 0.27$. However, the mixed condition did not differ from the control condition ($P = 0.676$), and the partisan sharing condition also did not significantly differ from the control condition ($P = 0.241$). Taken together, these results suggest that accuracy motivations increase truth discernment, but motivations to share articles that appeal to one's political in-group can decrease truth discernment.

We then examined how these motives influenced partisan bias. Replicating the results from experiment 1, there was a significant main effect of accuracy incentives on partisan bias, $F(1, 994) = 9.01$, $P = 0.003$, $\eta^2_G = 0.01$, but no effect of the partisan sharing manipulation, $F(1, 994) = 0.60$, $P = 0.441$, $\eta^2_G = 0.00$, and no interaction between the accuracy and the partisan sharing manipulation, $F(1, 994) = 0.27$, $P = 0.606$, $\eta^2_G = 0.00$. Post-hoc tests indicated that there was a non-significant difference in partisan bias between the accuracy incentives condition ($M = 1.26$, 95% CI 1.01–1.51) and the control condition ($M = 1.72$, 95% CI 1.47–1.98), $P = 0.062$, $d = 0.23$, a 27% decrease in partisan bias. There was a significant difference between the accuracy incentives condition and the partisan sharing condition ($M = 1.76$, 95% CI 1.48–2.03), $P = 0.040$, $d = 0.24$. No other post-hoc tests yielded significant differences (ps >0.182).

Follow-up analysis (Supplementary Appendix 1) once again indicated that the incentives primarily impacted the percentage of politically incongruent true headlines rated as accurate ($M = 55.61\%$, 95% CI 51.68–59.54) when compared with the control condition ($M = 37.65\%$, 95% CI 33.83–41.46), $P < 0.001$, $d = 0.58$. The incentives again did not impact congruent true news, incongruent false news or congruent false news (ps >0.148).

There was no significant effect of accuracy incentives on sharing discernment ($P = 0.996$), diverging from the results of study 1. However, follow-up analysis (Supplementary Appendix 1) indicated that those in the partisan sharing condition shared more politically congruent news (either true or false) ($M = 1.98$, 95% CI 1.90–2.05) as compared with the control condition ($M = 1.80$, 95% CI 1.74–1.87), $P = 0.015$, $d = 0.21$. Additionally, those in the mixed condition ($M = 2.02$, 95% CI 1.94–2.10) shared more politically congruent news (true or false) as compared with the control condition, $P < 0.001$, $d = 0.26$. Thus, prompting participants to identify whether an article will be liked by their political allies— whether or not they are also incentivized to be accurate—appears to increase intentions to share both true and false news that appeals to one's own partisan identity.

### Experiment 3: accuracy incentives and source cues

In experiment 3, we sought to replicate our prior findings in a nationally representative sample in the United States. We recruited a sample of 921 US participants that was quota matched to the national distribution on age, gender, ethnicity and political party. We also tested a potential psychological process underlying the effects of accuracy incentives. As prior work has found strong effects of source cues[17] on judgements of news headlines, we suspected that people were responding to source cues when making judgements about news. As true news often contains more recognizable sources with partisan connotations (for example, 'nytimes.com' as opposed to the fake news website 'yournewswire. com')[59], this may explain why incentives only impacted judgements of true news in experiments 1 and 2. To test this possibility, we examined the effect of incentives with and without source cues (for example, a URL name such as 'foxnews.com') present beside the headlines (for more details, see Methods). Because we wanted to compare the effects of accuracy incentives with and without sources, this study had four conditions: accuracy incentives (with sources), control (with sources), accuracy incentives (without sources) and control (without sources).

Replicating the main results from experiments 1 and 2, the accuracy incentives condition significantly improved truth discernment,

$F(1, 917) = 4.44$, $P = 0.035$, $\eta^2_G = 0.01$, reduced partisan bias, $F(1, 917) = 18.21$, $P < 0.001$, $\eta^2_G = 0.02$, and increased the number of politically incongruent true articles rated as accurate, $F(1, 917) = 20.94$, $P < 0.001$, $\eta^2_G = 0.02$. Thus, accuracy incentives appear to increase accuracy and reduce partisan bias in a large representative sample, suggesting that the results of these experiments probably generalize to the US population as a whole.

Although effect sizes appeared to be descriptively smaller when sources were removed from the headlines (for details, see Fig. 1 and Supplementary Appendix 1), we did not find significant interactions between the main outcome variables and the presence or absence of source cues. However, this study design did not provide strong power to test whether this was not due to chance, since interaction effects can require up to 16 times as much power as main effects[60,61] (for power analysis, see Methods). Additional analysis using Bayes factors[62] reported in Supplementary Appendix 1 did not find strong evidence for the absence of interaction effects. Like in experiment 2, there was once again no significant impact of accuracy incentives on sharing discernment ($P = 0.906$).

### Experiment 4: the effect of a non-financial intervention

In experiment 4, we replicated the accuracy incentive and control condition in another politically balanced sample of 983 US adults, but also added a non-financial accuracy motivation condition. This non-financial accuracy motivation condition was designed to rule out multiple interpretations behind our earlier findings. One mundane interpretation is that participants are merely saying what they believe fact-checkers think is true, rather than answering in accordance with their true beliefs. However, this non-financial intervention does not incentivize people to answer in ways that do not align with their actual beliefs. Additionally, because financial incentives are more difficult to scale to real-world contexts, the non-financial accuracy motivation condition speaks to the generalizability of these results to other, more scalable ways of motivating accuracy.

In the non-financial accuracy condition, people read a brief text about how most people value accuracy and how people think sharing inaccurate content hurts their reputation[63] (see intervention text in Supplementary Appendix 2). People were also told to be as accurate as possible and that they would receive feedback on how accurate they were at the end of the study.

Our main pre-registered hypothesis was that this non-financial accuracy motivation condition would increase belief in politically incongruent true news relative to the control condition. An analysis of variance (ANOVA) found a main effect of the experimental conditions on the amount of politically incongruent true news rated as true, $F(2, 980) = 17.53$, $P < 0.001$, $\eta^2_G = 0.04$. Supporting our main pre-registered hypothesis, the non-financial accuracy motivation condition increased the percentage of politically incongruent true news stories rated as true ($M = 43.97$, 95% CI 40.59–47.34) as compared with the control condition ($M = 35.19$, 95% CI 31.93–38.45), $P < 0.001$, $d = 0.29$. Replicating studies 1–3, the accuracy incentive condition also increased perceived accuracy of politically incongruent true news ($M = 49.15$, 95% CI 45.74–52.55), $P < 0.001$, $d = 0.45$. The accuracy incentive and non-financial accuracy motivation condition were not significantly different from one another ($P = 0.083$, $d = 0.17$), though this may be because we did not have enough power to detect a difference. In short, the non-financial accuracy motivation manipulation was also effective at increasing belief in politically incongruent true news, with an effect about 63% as large as the effect of the financial incentive.

Since we expected the non-financial accuracy motivation condition to have a smaller effect than the accuracy incentives condition, we did not pre-register hypotheses for truth discernment and partisan bias, as we did not anticipate having enough power to detect effects for these outcome variables. Indeed, the non-financial accuracy motivation condition did not significantly increase truth discernment

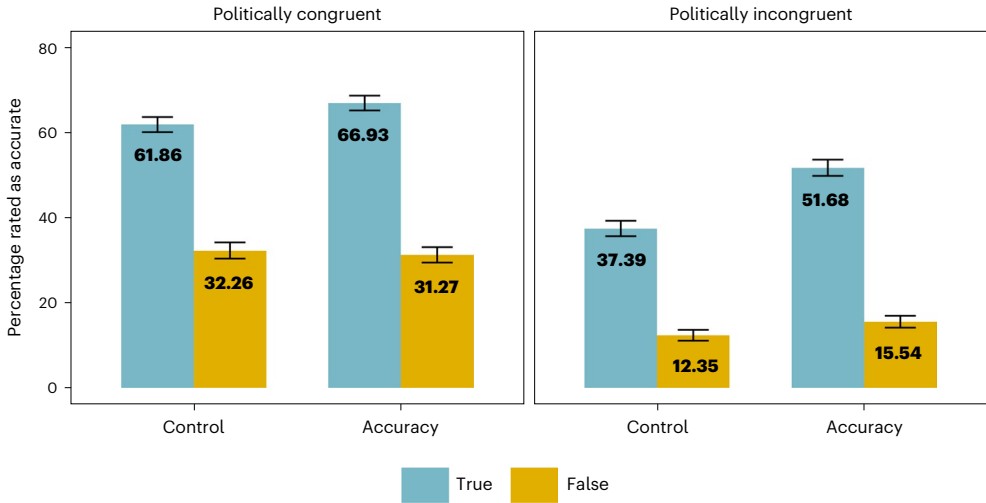

**Fig. 2 | Accuracy incentives had the strongest impact on belief in politically incongruent true news.** IDA results (with data from all four studies, n = 2,092) broken up by headline type. Incentives had the largest effect on belief in politically incongruent true news (d = 0.47), and a smaller effect on politically congruent true news (d = 0.17). Incentives did not have a significant effect on politically congruent or politically incongruent false news when controlling for multiple comparisons. Headline-level analysis revealed that incentives increased belief in all eight true items, but did not decrease belief in a single false item (for item-level analysis, see Supplementary Appendix 3). Means for each condition are shown in the figure, and error bars represent 95% confidence intervals.

($P$ = 0.221) or partisan bias ($P$ = 0.309). However, replicating studies 1–3, accuracy incentives once again improved truth discernment ($P$ = 0.001, $d$ = 0.28) and reduced partisan bias ($P$ = 0.003, $d$ = 0.25). The effect of the non-financial accuracy motivation condition was 47% as large as the effect of the accuracy incentive for truth discernment and 45% as large for partisan bias. There was also no overall effect of the experimental conditions on sharing discernment ($P$ = 0.689). For extended results, see Supplementary Appendix 1.

Together, these results suggest that a subtler (and also more scalable) accuracy motivation intervention that does not employ financial incentives is effective at increasing the perceived accuracy of true news from the opposing party, but has a smaller effect size than the stronger financial incentive intervention.

## IDA

To generate more precise estimates of our effects, we pooled data from all four studies to conduct an IDA[64]. For the IDA, we used only the 16 news headlines that were used in all four studies, and included only the accuracy incentives and control conditions that were used in all four studies.

We did not have any studies in the file drawer on this topic, meaning that our estimate was not influenced by publication bias.

Incentives had the largest positive effect on the perceived accuracy of politically incongruent true news, $P$ < .001, $d$ = 0.47; and a smaller positive effect on the perceived accuracy of politically congruent true news, $P$ = 0.001, $d$ = 0.17. Incentives did not significantly affect belief in politically incongruent false news, $P$ = 0.163, $d$ = 0.13, or belief in politically congruent false news, $P$ = 0.993, $d$ = −0.04 (Fig. 2), after adjusting for multiple comparisons with Tukey post-hoc tests. Analysis for each individual item revealed that incentives significantly increased belief in all true items, but they did not significantly decrease belief in any false items (though they significantly increased belief in one false item). More details are reported in Supplementary Appendix 1, and an analysis for each individual headline is reported in Supplementary Appendix 3. Additional analysis using Bayes factors reported in Supplementary Appendix 4 found strong evidence that incentives impacted belief in both politically congruent and politically incongruent true news, but found inconsistent evidence that they affected belief in false news.

While effects on sharing discernment were inconsistent across studies, the IDA found that there was a small positive effect of the

incentive on sharing discernment, $t$(2020.20) = 2.19, $P$ = 0.029, $d$ = 0.10. Finally, people spent slightly more time on each headline in the accuracy incentives condition, $t$(818.53) = 2.34, $P$ = 0.019, $d$ = 0.16, indicating that incentives may have led people to put more effort into their responses.

Replicating prior work[26–31], conservatives were worse at discerning between true and false headlines than liberals. Conservatives answered about 9.26 (out of 16) questions correctly when not incentivized to be accurate, and liberals answered 10.93 questions out of 16 correctly when unincentivized – a 1.67-point difference, 95% CI 1.41–1.94, $t$(1035.69) = 12.53, $P$ < 0.001, $d$ = 0.77. However, when conservatives were incentivized to be accurate, they answered 10.12 questions correctly, making the gap between incentivized conservatives and unincentivized liberals 0.81 points, 95% CI 0.53–1.09, $t$(951.91) = 5.65, $P$ < 0.001, $d$ = 0.35. In other words, paying conservatives less than a dollar to correctly identify news headlines as true or false reduced the gap in performance between conservatives and (unincentivized) liberals by 51.50%. Incentives also considerably reduced the gap between conservatives and liberals in terms of partisan bias, sharing discernment and belief in politically incongruent true news. More detail is reported in Supplementary Appendix 1 and plotted visually in Fig. 3. Altogether, these results suggest that a substantial portion of US conservatives' tendency to believe and share less accurate news reflects a lack of motivation to be accurate rather than lack of knowledge alone.

Importantly, the incentives improved truth discernment for both liberals, $d$ = 0.23, $P$ < 0.001, and conservatives, $d$ = 0.40, $P$ < 0.001 (for table of effect sizes broken down by political affiliation, see Supplementary Appendix 5). Descriptively, the effect sizes for our intervention were larger for conservatives than liberals, which diverges from other misinformation interventions that tend to show larger effect sizes for liberals[65,66]. Furthermore, political ideology (liberal versus conservative) was a significant moderator of belief in incongruent true news, $P$ = 0.033, and partisan bias, $P$ = 0.029 (though this moderation effects was not significant for truth discernment, $P$ = 0.095, or sharing discernment, $P$ = 0.061), such that the effects of incentives appeared to be larger for conservatives than liberals. The effect of the incentives on truth discernment was not significantly moderated by cognitive reflection, political knowledge or affective polarization (ps <0.182). However, even though we had a large sample, we were still slightly underpowered to detect these interaction effects (see power analysis in Methods), and

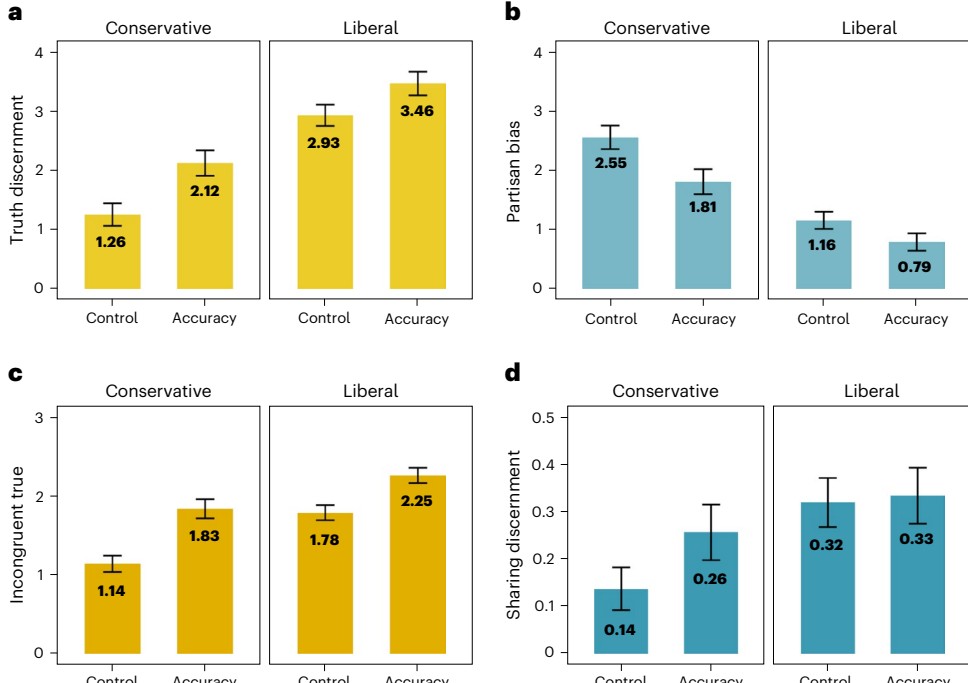

**Fig. 3 | Accuracy incentives closed the gap between conservatives and liberals for several outcome variables. a**, Conservatives were worse at truth discernment as compared with liberals. **b**–**d**, They also showed more partisan bias (**b**), less belief in politically incongruent true news (**c**) and worse sharing discernment (**d**). However, incentives consistently closed the gap between conservatives and (unincentivized) liberals for all of these outcome variables, suggesting that conservatives' greater tendency to believe in and share (mis) information may in part reflect a lack of motivation to be accurate (instead of lack of knowledge or ability alone). The data shown are the pooled data across all four studies ($n = 2,092$). Means for conservatives and liberals are shown in the figures, and all error bars represent 95% confidence intervals. The $y$ axis for each graph represents (**a**) truth discernment, (**b**) partisan bias, (**c**) belief in politically incongruent true news and (**d**) sharing discernment.

supplemental Bayesian analyses also did not find strong evidence for the significant moderation effects (Supplementary Appendix 11), so these interaction effects should be interpreted with caution.

**Relative importance of accuracy incentives**
In each experiment, we measured other individual difference variables known to be predictive of truth discernment, such as cognitive reflection, political knowledge and partisan animosity, as well as demographic variables, such as age, education and gender. We ran a multiple regression analysis on our IDA with all of these variables included in the model (Fig. 4a). To compare the relative importance of each of these predictors, we also ran a relative importance analysis using the 'lmg' method[67], which calculates the relative contribution of each predictor to the $R^2$ (Fig. 4b). Full models and relative importance analyses are in Supplementary Appendix 6 and 7.

Political conservatism and accuracy incentives were among the most important predictors for many of the key outcome variables, although confidence intervals were large and overlapping for the relative importance analysis (Supplementary Appendix 4). While prominent accounts claim that partisanship and politically motivated cognition play a limited role in the belief and sharing of misinformation as compared with other factors (such as cognition reflection or inattention)[10,68], our results indicate that motivation and partisan identity or ideology are very important factors. Our data point to the importance of broad theoretical accounts of (mis)information belief and sharing that integrate motivation and partisan identity with other variables[2,10,11,24,69]. Indeed, an investigation using cognitive modelling found that a broad model of misinformation belief that included multiple factors (such as partisan identity, cognitive reflection and more) performed better at predicting acceptance of misinformation than other models that focused exclusively on cognitive or emotional factors[70].

## Discussion
Increasing motivations to be accurate via a small financial incentive improved people's accuracy in discerning between true and false news and decreased the partisan divide in belief in news by about 30%. These effects were observed across four experiments ($n = 3,364$), and were primarily driven by an increase in the perceived accuracy of politically incongruent true news ($d = 0.47$). No significant effects were found for false news, which people encounter relatively infrequently online[71]. Additionally, providing people with an incentive to identify articles that would be liked by their political in-group reduced accuracy and increased intentions to share politically congruent true and false news. Thus, social or partisan identity goals appear to interfere with accuracy goals. Furthermore, a non-financial accuracy motivation intervention that provided people feedback about their accuracy, emphasized social norms about accuracy and highlighted the reputational benefits of being accurate significantly increased the perceived accuracy of politically incongruent true news ($d = 0.29$). This illustrates that accuracy motivation interventions that do not involve financial incentives can be applied at scale.

These results make a number of key theoretical contributions. First, they suggest that partisan differences in news judgements do not simply reflect differences in factual knowledge[10]. Instead, our data suggest that a substantial portion of this partisan divide can be attributed to a lack of motivation to be accurate. While there have been debates about whether partisan differences in belief reflect differing prior beliefs versus politically motivated cognition[21,22], our studies provide robust causal evidence for the effect of motivation on belief.

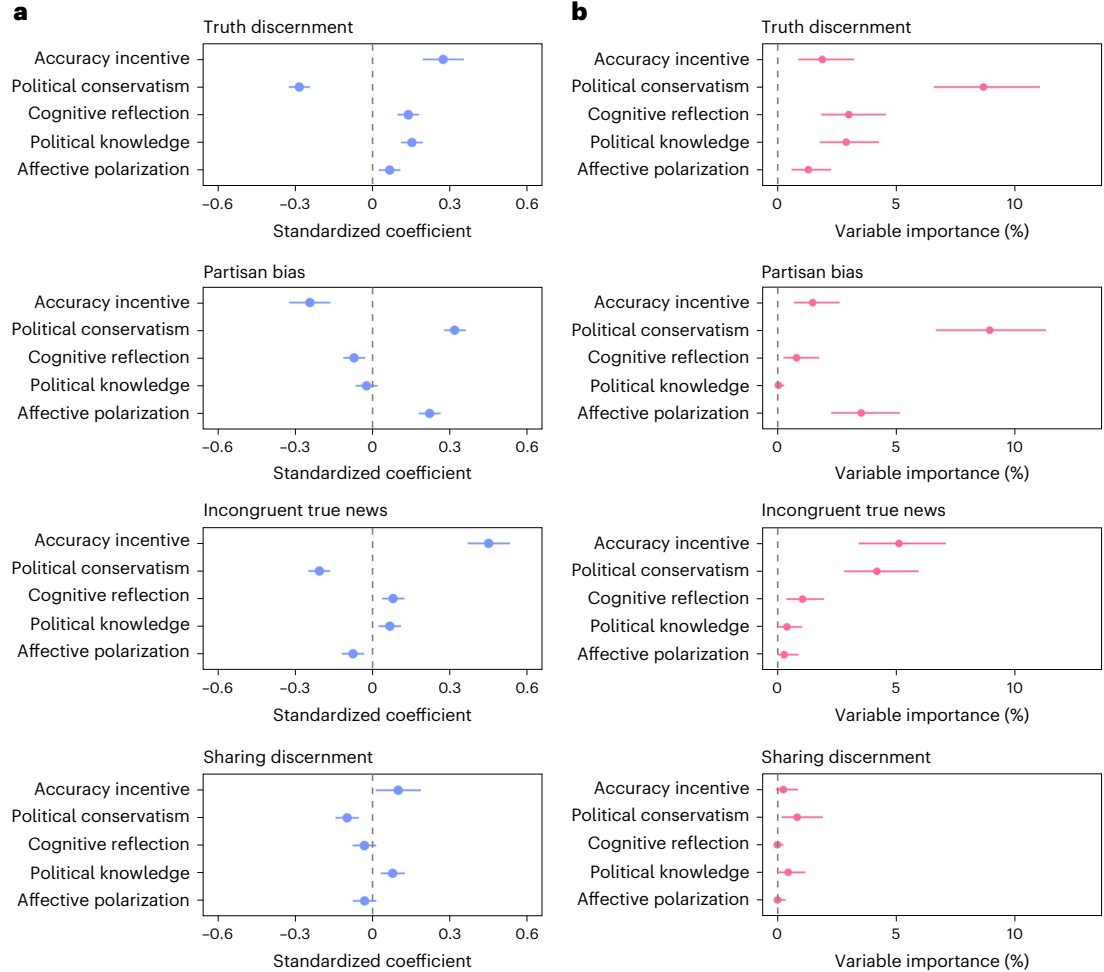

**Fig. 4 | Relative importance of accuracy incentives. a**, Multiple regression results for the main outcome variables: truth discernment, partisan bias, belief in incongruent true news, and sharing discernment. Standardized beta coefficients are plotted for ease of interpretation. **b**, Variable importance estimates (lmg values) with bootstrapped confidence intervals are shown to examine the estimated percentage contribution of each predictor to the $R^2$. The data shown are the pooled data across all four studies ($n = 2,092$). All error bars represent 95% confidence intervals.

Additionally, while a number of studies have observed that American conservatives tend to be more susceptible to misinformation than liberals[26–31], incentives closed the gap in truth discernment between liberals and (unincentivized) conservatives by more than half. This suggests that a significant portion of partisan differences in (mis) information belief can be attributed to motivational factors, rather than reflecting knowledge gaps alone.

Along with other research[39,72,73], these findings suggest that survey data about belief in (mis)information should not be taken at face value. People respond differently when they are highly motivated to be accurate compared with when they are motivated to appeal to their in-group[50]. However, this does not mean that prior beliefs are not also important, or that motivation is relevant in every context. Indeed, judgements of false headlines appeared to be unaffected by accuracy motivations, suggesting that other factors may play a more prominent role in people's assessment of false news as compared with true news. Future work can explore why incentives have different effects for true and false news. However, since people encounter fake news websites rarely, some have argued that it is more important to increase trust in reliable news than decrease belief in false news[74].

These results also have practical implications for interventions for improving the accuracy of people's beliefs and sharing decisions[75,76]. Accuracy incentives improved the accuracy of people's judgements,

and an IDA found that this effect may have spilled over into intentions to share more accurate articles (though this effect was small and inconsistent across studies). Further, making people think about what headlines would be liked by their in-group increased people's intentions to share both politically congruent false (and true) news. Thus, interventions and social media design features should aim to both increase accuracy motivations and decrease motivations to share content that receives high social reward at the cost of accuracy. In line with this, experimental studies have found that providing social rewards for sharing high-quality content and punishments for sharing low-quality content[77] improves the quality of news people report intending to share. Additionally, making people publicly endorse that the news that they share is accurate[78], or showing people that fellow in-group members believe content is misleading[79], also improves people's sharing intentions. Future work should continue to explore how to incentivize people to engage with more accurate content online by, for example, emphasizing social norms around accuracy or emphasizing the reputational benefits of sharing accurate content (as in experiment 4).

One limitation of this work is that survey experiments have unknown ecological validity. To maximize ecological validity, we used real, pre-tested news headlines in the format in which they would be regularly encountered on social media websites such as Facebook. Additionally, self-reported sharing intentions are highly correlated

with real online news sharing[80], and a field experiment suggests that priming accuracy can improve news sharing decisions on Twitter[68], illustrating that results from survey experiments on misinformation can translate to the field. Another potential limitation is that there are multiple ways to interpret the effects of financial incentives. For instance, people may be guessing what they think fact-checkers believe to earn money, rather than expressing their true beliefs. However, this interpretation is unlikely to explain the full effect, since a subtle non-financial accuracy motivation intervention had similar (albeit smaller) effects. Furthermore, supplementary analysis found that few participants reported answering in ways that did not accord with their true beliefs to receive money (Supplementary Appendix 1).

## Conclusions

There is a sizeable partisan divide in the kind of news liberals and conservatives believe in, and conservatives tend to believe in and share more false news than liberals. Our research suggests these differences are not immutable. Motivating people to be accurate improves accuracy about the veracity of true (but not false) news headlines, reduces partisan bias and closes a substantial portion of the gap in accuracy between liberals and conservatives. Theoretically, these results identify accuracy and social motivations as key factors in driving news belief and sharing. Practically, these results suggest that shifting motivations may be a useful strategy for creating a shared reality across the political spectrum.

## Methods

We report how we determined our sample size, all data exclusions, all manipulations and all measures in the experiment. The research methods were approved by the University of Cambridge Psychology Ethics Committee (Protocol #PRE.2020.110). These studies were pre-registered. Stimuli, Qualtrics survey files, anonymized data, analysis code and all pre-registrations are available on our Open Science Framework (OSF) page: https://osf.io/75sqf.

### Experiment 1

**Participants.** The experiment launched on 30 November 2020. We recruited 500 participants via the survey platform Prolific Academic[55]. Specifically, we recruited 250 conservative participants and 250 liberal participants from the United States via Prolific Academic's demographic pre-screening service to ensure the sample was politically balanced. Our a priori power analysis indicated that we would need 210 participants to detect a medium effect size of $d = 0.50$ at 95% power, though we doubled this sample size to account for partisan differences and oversampled to account for exclusions. A total of 511 participants took our survey. Following our pre-registered exclusion criteria, we excluded 32 participants who failed our attention check (or did not get far enough in the experiment to reach our attention check), and an additional 17 participants who said they responded randomly at some time during the experiment. This left us with a total of 462 participants (194 M, 255 F, 12 trans/non-binary; age: $M = 35.85$, standard deviation (s.d.) 13.66; politics: 253 Democrats, 201 Republicans). The experiment 1 pre-registration is available at https://aspredicted.org/blind.php?x=gk9xg5.

**Materials.** The materials were 16 pre-tested true and false news headlines from a large pre-tested sample of 225 news headlines[56]. In total, eight of these news headlines were false, and eight of the news headlines were true. Because we were interested in whether accuracy incentives would reduce partisan bias, we specifically selected headlines that had a sizeable gap in perceived accuracy between Republicans and Democrats as reported in the pre-test, as well as headlines that were not outdated (the pre-test was conducted a few months before the first experiment). Specifically, we chose eight headlines (four false and four true) that Democrats rated as more accurate than Republicans in the pre-test, and eight headlines (four false and four true) that Republicans rated as more accurate than Democrats. For example stimuli, see Supplementary Appendix 8, and for full materials, see the OSF page.

### Procedure

**News evaluation task.** Participants were shown these 16 news headlines, along with an accompanying picture and source (similar to how a news article preview would show up on someone's Facebook feed), and asked 'To the best of your knowledge, is the claim in the above headline accurate?' on a scale from 1 ('extremely inaccurate') to 6 ('extremely accurate'). Afterwards, they were asked 'If you were to see the above article on social media, how likely would you be to share it?' on a scale from 1 ('extremely unlikely') to 6 ('extremely likely').

**Accuracy incentives manipulation.** Half of the participants were randomly assigned to a control condition, in which we explained the news evaluation task, but we did not provide any information about a bonus payment. The other half were assigned to an accuracy incentives condition. In this condition, we explained the news evaluation task, and then told participants they would receive a 'bonus payment of up to $1.00 based on how many correct answers [they] provide regarding the accuracy of the articles. Correct answers are based on the expert evaluations of non-partisan fact-checkers.' Specifically, they received one dollar for answering 15 out of 16 questions correctly, and 50 cents for answering 13 out of 16 questions correctly. Since we measured accuracy on a continuous scale, we told participants that 'if the headline describes a true event, either 'slightly accurate', 'moderately accurate' or 'extremely accurate' constitute correct responses. Similarly, if the headline describes a false event, either 'extremely inaccurate', 'moderately inaccurate' or 'slightly inaccurate' constitute 'correct' responses. In other words, the continuous scale was measured dichotomously for the purposes of giving financial incentives. Participants were also notified that all other questions would not affect their bonus payment. For full manipulation text, see Supplementary Materials 2 or the OSF.

**Other measures.** We gave participants a three-item cognitive reflection task[52]. We measured participants' political knowledge using a five-item scale[49] and in-group love/out-group hate with feeling thermometers[81]. For question text, see Supplementary Appendix 9 and the OSF. These measures were repeated across all studies.

**Analysis.** For truth discernment, partisan bias and sharing discernment, two-sided independent samples $t$-tests were used. While we asked participants to rate the truth of headlines on a continuous scale, these variables were recoded as dichotomous for analysis because the financial incentive only rewarded participants on the basis of whether they correctly identified a headline as true or false. Since we did not clearly specify this in the experiment 1 pre-registration (but did for experiments 2–4), we show the results with a continuous coding in Supplementary Appendix 10. The continuous coding did not change the conclusions of our studies.

To test what types of headlines were affected by the incentives, we ran a 2 (accuracy incentive versus no incentive) × 2 (politically congruent versus politically incongruent) × 2 (true headlines versus false headlines) mixed-design ANOVA with the percentage of articles rated as accurate as the dependent variable, and then followed up with Tukey HSD post-hoc tests. Extended analyses are in Supplementary Appendix 1.

### Experiment 2

**Participants.** The experiment launched on 22 January 2021. We aimed to recruit 1,000 total participants (250 per condition) via the survey platform Prolific Academic, though we oversampled and recruited 1,100 to account for exclusion criteria. We chose this sample size because a power analysis revealed that we needed at least 216 participants per

condition to detect the smallest effect size ($d = 0.24$) at 0.80% power using a one-tailed $t$-test (although two-tailed tests were used for all analysis). Once again, we used Prolific's pre-screening platform to recruit 550 liberals and 550 conservatives from the United States, and 1,113 participants took our survey. Following our pre-registered exclusion criteria, we excluded 76 participants who failed our attention check (or did not finish enough of the survey to reach the attention check) and an additional 39 participants who said they responded randomly at some point during the experiment. This left us with a total of 998 participants in total (463 M, 505 F, 30 transgender/non-binary/other; age: $M = 36.17$, s.d. 13.94; politics: 568 liberals, 430 conservatives). This experiment was also pre-registered (pre-registration at https://asprediced.org/blind.php?x=/FKF_15L).

**Partisan sharing and mixed incentives manipulations.** In the new partisan sharing condition, participants were first asked before the experiment to report the political party with which they identify. Then, they were told that they would receive a bonus payment of up to $1.00 based on how accurately they identified information that would be liked by members of their political party if they shared it on social media. Bonuses were awarded on the basis of how closely participants' answers matched partisan alignment scores from a pre-test[48]. Before each question about accuracy and sharing, participants were asked 'If you shared this article on social media, how likely is it that it would receive a positive reaction from [your political party] (for example, likes, shares, and positive comments)?' In the mixed condition, participants were first given financial incentives for both correctly identifying whether the article would be liked by a member of their political party, and were then asked about accuracy and given incentives for identifying whether the article was accurate. For full intervention text, see Supplementary Appendix 2.

**Analysis.** To understand the impact of accuracy and partisan sharing motivations on truth discernment and partisan bias, we ran 2 (accuracy incentive versus control) × 2 (partisan sharing versus control) ANOVAs and followed up on the results using Tukey HSD post-hoc tests. To test what types of headlines were affected by the incentives, we ran a 2 (accuracy versus control) × 2 (partisan sharing versus control) × 2 (politically congruent versus politically incongruent) × 2 (true headlines versus false headlines) mixed-design ANOVA with the percentage of articles rated as accurate as the dependent variable, and then followed up with Tukey HSD post-hoc tests.

### Experiment 3
**Participants.** The experiment launched on 13 June 2021. We aimed to recruit a nationally representative sample (quota matched to the US population distribution by age, ethnicity and gender) of 1,000 participants via the survey platform Prolific. As in studies 1 and 2, we ensured that the nationally representative sample was politically balanced, or half liberal and half conservative. A total of 1,055 total participants took the survey. Then, we once again excluded 95 participants who failed our attention check (or did not make it to that point in the survey), as well as 39 participants who said they were responding randomly at some point in the survey. This left us with a total of 921 participants (439 M, 470 F, 12 transgender/non-binary/other; age: $M = 40.07$, s.d. 14.67; politics: 542 liberals, 379 conservatives). This experiment was also pre-registered (pre-registration available at https://aspredicted.org/7M2_9K9).

**Materials.** We once again used the same 16 pre-tested true and false news headlines in addition to eight extra true and false news items from the same pre-test. For consistency, we report the results of the 16 news items in the paper, but we also report the results for the full set of 24 items in Supplementary Appendix 3, which did not change our conclusions.

**Manipulations.** In addition to the accuracy incentive and control condition, participants were assigned to identical accuracy incentive and control conditions without source cues present on the stimuli. In these conditions, the sources (for example, 'nytimes.com') were greyed out, so participants could only make assessments of the stimuli based on the photo and headline alone (for examples, see Supplementary Materials 8).

**Analysis.** To understand the impact of accuracy incentives and source cues on truth discernment and partisan bias, we ran 2 (accuracy versus control) × 2 (source versus no source) ANOVAs and followed up on the results using Tukey HSD post-hoc tests. To test what types of headlines were affected by the incentives, we ran a 2 (accuracy versus control) × 2 (source versus no source) × 2 (politically congruent versus politically incongruent) × 2 (true headlines versus false headlines) mixed-design ANOVA with the percentage of articles rated as accurate as the dependent variable, and then followed up with Tukey HSD post-hoc tests.

**Power analysis for interaction effects.** On the basis of the effect sizes of study 2 and the principle that 16 times the sample size is needed to detect an attenuated interaction effect[60,61], a power analyses conducted after we ran the study found that we needed roughly 1,536 participants to detect an interaction for the amount of politically incongruent news rated as true, 2,560 participants to detect an interaction effect for truth discernment and 7,488 participants to detect an interaction effect for partisan bias with 80% power. Thus, this particular design was underpowered to detect whether accuracy incentives interacted with source cues.

### Experiment 4
**Participants.** This experiment launched on 25 May 2022. We aimed to recruit a total of 1,000 participants (roughly 333 per condition) via the platform Prolific academic. We chose this sample size as a power analysis found that we would need 312 per condition to detect the smallest effect size found in the previous study ($d = 0.26$) with 90% power. Additionally, we wanted relatively high power because we expected the effect of the non-financial accuracy motivation condition to be smaller than that of the financial incentive condition. We used Prolific's pre-screening platform to recruit a sample that was balanced by politics and gender. A total of 1,007 participants took our survey. Following our pre-registered exclusion criteria, we excluded 17 participants who failed our attention check (or did not finish enough of the survey to reach the attention check) and an additional 8 participants who said they responded randomly at some point during the experiment. This left us with a total of 993 participants in total (486 M, 483 F, 30 transgender/non-binary/other; age: $M = 41.46$, s.d. 15.06; politics: 507 liberals, 476 conservatives). This experiment was also pre-registered (pre-registration available at https://aspredicted.org/86W_BY4).

**Materials.** We once again used the same 16 pre-tested true and false news headlines extra 'misleading' news headlines.

**Analysis.** Following our pre-registered analysis plan, we ran a one-way (accuracy versus control versus non-financial accuracy motivation) ANOVA with the percentage of incongruent-true articles rated as true as the dependent variable, followed up by Tukey post-hoc tests. We also ran one-way ANOVAs with truth discernment and partisan bias dependent variables and followed up with post-hoc tests.

### IDA
**Analysis.** We conducted moderation analysis on the pooled dataset by testing for an interaction between the condition and political ideology (liberal versus conservative) in a linear regression. To test the relative importance of each predictor, we ran a relative importance analysis using the 'reliampo' package in R. Bootstrapped confidence intervals were calculated for 'lmg' variables using 1,000 bootstraps.

**Power analysis for moderation effects.** Using effect sizes from the IDA and the principle that 16 times the sample size is needed to detect an attenuated interaction effect[60,61], a post-hoc power analysis found that we needed 2,336 participants to detect an interaction effect for the amount of politically incongruent news rated as true, 5,984 participants to detect an interaction effect for truth discernment, 7,488 for partisan bias and 50,336 to detect an interaction for sharing discernment. Thus, moderation effects should be interpreted with caution.

**Signal detection analysis.** As another robustness check, we also conducted supplemental analysis using signal detection modelling[15]. This analysis found that incentives increased participants' discrimination between true and false news (for both politically congruent and politically incongruent headlines), and also increased the threshold by which people accepted politically incongruent headlines as true (Supplementary Appendix 12). In sum, analysis using signal detection modelling yielded highly similar results to our main analysis.

### Reporting summary

Further information on research design is available in the Nature Portfolio Reporting Summary linked to this article.

## Data availability

Anonymized data, Qualtrics files and stimuli are available on the OSF at https://osf.io/75sqf.

## Code availability

The R code necessary to reproduce our results is available on the OSF at https://osf.io/75sqf.

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

## Acknowledgements

We are grateful for support from a Gates Cambridge Scholarship awarded to S.R. (grant #OPP1144), a British Academy Postdoctoral Fellowship awarded to J.R. (#PF21\210010), a John Templeton Foundation Grant (#61378) awarded to J.J.V.B., a Russell Sage Foundation Grant awarded to S.R. and JV.B., and an Infodemic grant awarded to S.V.L. (UK Government, #SCH-00001-3391). The funders had no role in study design, data collection and analysis, decision to publish or preparation of the paper.

## Author contributions

S.R., J.R., S.V.L. and J.J.V.B. designed the experiments. S.R. implemented the study design and collected the data. S.R. and J.R. analysed the data. S.R. drafted the paper. All authors provided critical revisions and approved the final paper for submission.

## Competing interests

The authors declare no competing interests.

## Additional information

**Correspondence and requests for materials** should be addressed to Steve Rathje or Sander van der Linden.

# Reporting Summary

## Statistics

For all statistical analyses, confirm that the following items are present in the figure legend, table legend, main text, or Methods section.

| n/a | Confirmed | |
|---|---|---|
| ☐ | ☒ | The exact sample size (*n*) for each experimental group/condition, given as a discrete number and unit of measurement |
| ☐ | ☒ | A statement on whether measurements were taken from distinct samples or whether the same sample was measured repeatedly |
| ☐ | ☒ | The statistical test(s) used AND whether they are one- or two-sided<br>*Only common tests should be described solely by name; describe more complex techniques in the Methods section.* |
| ☐ | ☒ | A description of all covariates tested |
| ☐ | ☒ | A description of any assumptions or corrections, such as tests of normality and adjustment for multiple comparisons |
| ☐ | ☒ | A full description of the statistical parameters including central tendency (e.g. means) or other basic estimates (e.g. regression coefficient) AND variation (e.g. standard deviation) or associated estimates of uncertainty (e.g. confidence intervals) |
| ☐ | ☒ | For null hypothesis testing, the test statistic (e.g. *F*, *t*, *r*) with confidence intervals, effect sizes, degrees of freedom and *P* value noted<br>*Give P values as exact values whenever suitable.* |
| ☐ | ☒ | For Bayesian analysis, information on the choice of priors and Markov chain Monte Carlo settings |
| ☒ | ☐ | For hierarchical and complex designs, identification of the appropriate level for tests and full reporting of outcomes |
| ☐ | ☒ | Estimates of effect sizes (e.g. Cohen's *d*, Pearson's *r*), indicating how they were calculated |

*Our web collection on statistics for biologists contains articles on many of the points above.*

## Software and code

Policy information about availability of computer code

| Data collection | All data was collected via the survey platform Qualtrics in 2020-2021 (Version 2020 and 2021). |
|---|---|
| Data analysis | All data were analyzed using using R version 4.01. A number of R packages were used for analysis, such as rstatix (stats and effect sizes, version 0.6.0), relampo (for relative importance analysis, verison 2.2.5), ggplot2 (for plotting, version 3.4.0), BayesFactor (for Bayesian analysis, version 0.9.12.3), and jtools (for regression, version 2.1.4). Analysis code is available at our OSF: https://osf.io/75sqf. |

For manuscripts utilizing custom algorithms or software that are central to the research but not yet described in published literature, software must be made available to editors and reviewers. We strongly encourage code deposition in a community repository (e.g. GitHub). See the Nature Portfolio guidelines for submitting code & software for further information.

## Data

Policy information about availability of data

All manuscripts must include a data availability statement. This statement should provide the following information, where applicable:
- Accession codes, unique identifiers, or web links for publicly available datasets
- A description of any restrictions on data availability
- For clinical datasets or third party data, please ensure that the statement adheres to our policy

Anonymized data, Qualtrics files, and stimuli are available on the Open Science Framework (OSF): https://osf.io/75sqf.

## Human research participants

Policy information about studies involving human research participants and Sex and Gender in Research.

| | |
|---|---|
| Reporting on sex and gender | All participants reported their gender, but not their biological sex, as it was not considered essential to this analysis. was not considered essential Participants were given the opportunity to identify as male, female, transgender male, transgender female, non-binary, or other. In other multiple regression model, we included gender as a covariate, along with other demographic covariates, such as age and political affiliation. We had no prior hypotheses about the role of sex and gender in our research questions. |
| Population characteristics | See above |
| Recruitment | For Study 1, 2, and 4, the samples were convenience samples collected via the survey platform Prolific Academic. For Study 3, the sample was a nationally representative sample collected via the survey platform Prolific Academic. Since the data was collected via Prolific, it could have self-selection biases informed by the users of Prolific. For instance, it may over-representative of internet-savvy researchers who know about Prolific and are interested in taking surveys. Our survey with a nationally representative sample (Study 3) helps protect against these confounds. |
| Ethics oversight | The research methods were approved by the University of Cambridge Psychology Ethics Committee (Protocol #PRE.2020.110). |

Note that full information on the approval of the study protocol must also be provided in the manuscript.

# Field-specific reporting

Please select the one below that is the best fit for your research. If you are not sure, read the appropriate sections before making your selection.

☐ Life sciences    ☒ Behavioural & social sciences    ☐ Ecological, evolutionary & environmental sciences

For a reference copy of the document with all sections, see nature.com/documents/nr-reporting-summary-flat.pdf

# Behavioural & social sciences study design

All studies must disclose on these points even when the disclosure is negative.

| | |
|---|---|
| Study description | Four experiments investigating whether motivating people to be accurate (for example, via financial incentives) changes perceptions of true and false news. |
| Research sample | Studies 1, 2, and 4 have politically-balanced samples recruited via Prolific Academic (e.g., half-Democrat, half-Republican). For Study 3, a nationally-representative (quota-matched to the US distribution of age, gender, political party, and race/ethnicity) sample was collected via Prolific Academic. |
| Sampling strategy | Data was collected via the survey platform Qualtrics, and the sample was collected via Prolific. For Study 1, 2, and 4, the samples were convenience samples collected via the survey platform Prolific Academic. For Study 3, the sample was a nationally representative sample collected via the survey platform Prolific Academic. |
| Data collection | Data was collected via the survey platform Qualtrics. Because all randomization happened automatically through Qualtrics, the researcher was effectively blind to the study condition when collecting data. |
| Timing | Study 1 data was collected on Nov. 30, 2020, Study 2 data was collected on Jan. 22, 2021, and Study 3 data was collected on June 13, 2021. Stopping times were determined by when Prolific Academic hit our target sample size. |
| Data exclusions | l exclusion criteria were preregistered. In Study 1, 32 participants were excluded for failing an attention check at the end of the survey (or not getting to that point in the survey). An additional 17 participants were also excluded from Study 1 for reporting responding randomly at any time during the experiment. In Study 2, 76 participants were excluded for failing our attention check/not completing the survey, and an additional 39 participants were excluded for reporting responding randomly at some point during the survey. In Study 3, 95 participants were excluded for failing the attention check/not completing the survey, and an additional 39 participants were excluded for reporting responding randomly at some point during the survey. In Study 4, we excluded 16 participants who failed our attention check (or did not finish enough of the survey to reach the attention check) and an additional 8 participants who said they responded randomly at some point during the experiment. |
| Non-participation | Study 1, 10 participants did not complete the survey. In Study 2, 14 participants did not complete the survey. In Study 3, 40 participants did not complete the survey. In Study 4, 6 participants did not complete the survey. These participants were not included for analysis as they were captured by our exclusion criteria. |
| Randomization | In all studies, participants were randomized to experimental condition via Qualtric's randomization feature. |

# Reporting for specific materials, systems and methods

We require information from authors about some types of materials, experimental systems and methods used in many studies. Here, indicate whether each material, system or method listed is relevant to your study. If you are not sure if a list item applies to your research, read the appropriate section before selecting a response.

## Materials & experimental systems

| n/a | Involved in the study |
|-----|----------------------|
| ☒ ☐ | Antibodies |
| ☒ ☐ | Eukaryotic cell lines |
| ☒ ☐ | Palaeontology and archaeology |
| ☒ ☐ | Animals and other organisms |
| ☒ ☐ | Clinical data |
| ☒ ☐ | Dual use research of concern |

## Methods

| n/a | Involved in the study |
|-----|----------------------|
| ☒ ☐ | ChIP-seq |
| ☒ ☐ | Flow cytometry |
| ☒ ☐ | MRI-based neuroimaging |

