## [Peer Review File · Nature Human Behaviour]

Peer Review Information

Journal: Nature Human Behaviour

Manuscript Title: Accuracy and Social Motivations Shape Judgements of (Mis)Information

Corresponding author name(s): Steve Rathje and Sander van der Linden

Reviewer Comments & Decisions:

Decision Letter, initial version:

4th April 2022

Dear Mr Rathje,

Thank you once again for your manuscript, entitled "Accuracy and Social Incentives Shape Belief in (Mis)Information", and for your patience during the peer review process.

Your Article has now been evaluated by 4 referees. You will see from their comments copied below that, although they find your work of potential interest, they have raised quite substantial concerns. In light of these comments, we cannot accept the manuscript for publication, but would be interested in considering a revised version if you are willing and able to fully address reviewer and editorial concerns.

We hope you will find the referees' comments useful as you decide how to proceed. If you wish to submit a substantially revised manuscript, please bear in mind that we will be reluctant to approach the referees again in the absence of major revisions. We are committed to providing a fair and constructive peer-review process. Do not hesitate to contact us if there are specific requests from the reviewers that you believe are technically impossible or unlikely to yield a meaningful outcome.

To guide the scope of the revisions, the editors discuss the referee reports in detail within the team, including with the chief editor, with a view to (1) identifying key priorities that should be addressed in revision and (2) overruling referee requests that are deemed beyond the scope of the current study. We hope that you will find the prioritised set of referee points to be useful when revising your study. Please do not hesitate to get in touch if you would like to discuss these issues further.

1. Reviewers 2 and 4 raise important concerns about alternative explanations and Reviewers 1 and 2 question the ecological validity of the findings. Additionally, Reviewers 1 and Reviewer 4 ask that you clearly motivate your research and place your work within the context of the existing literature. Reviewer 2 raises major concerns about the scientific advance and the construal of existing literature.

We ask that you fully address these comments in revision.

2. Reviewer 1 and Reviewer 4 raise important concerns about the robustness of the evidence and the reporting of the results. Reviewer 4 raises important issues about power to test H4 and about the lack of appropriate statistical tests to interpret null results. In revision, please use Bayes Factors or equivalence testing to provide evidence in support of null hypotheses. Please refer to the following publication for the appropriate use of Bayes Factors: <https://www.ejwagenmakers.com/2015/BayesianAnalysisEnclopedia.pdf>. Additionally, please include sensitivity analyses to demonstrate that your design is sufficiently powered to detect the interaction effect sizes of interest.

If you wish to submit a suitably revised manuscript we would hope to receive it within 4 months. I would be grateful if you could contact us as soon as possible if you foresee difficulties with meeting this target resubmission date.

- Include a "Response to the editors and reviewers" document detailing, point-by-point, how you addressed each editor and referee comment. If no action was taken to address a point, you must provide a compelling argument. When formatting this document, please respond to each reviewer comment individually, including the full text of the reviewer comment verbatim followed by your response to the individual point. This response will be used by the editors to evaluate your revision and sent back to the reviewers along with the revised manuscript.
- Highlight all changes made to your manuscript or provide us with a version that tracks changes.

[REDACTED]

Thank you for the opportunity to review your work. Please do not hesitate to contact me if you have any questions or would like to discuss the required revisions further.

Sincerely,

Samantha Antusch

Samantha Antusch, PhD

Editor
Nature Human Behaviour

Reviewer expertise:

Reviewer #1: misinformation/disinformation; accuracy-increasing interventions and interventions to combat misinformation

Reviewer #2: partisanship; misinformation belief / sharing

Reviewer #3: psychology; fake news / misinformation

Reviewer #4: political science / political psychology; partisanship and belief in misinformation / misinformation sharing; accuracy-increasing interventions / interventions to combat misinformation

REVIEWER COMMENTS:

Reviewer #1:
Remarks to the Author:

The authors are to be commended for choosing a topic of such high importance and social relevance and collecting a large sample size for three experiments. I'm generally satisfied with the authors' approach, and I think the research findings provide significant implications to the current misinformation literature. Below I have some comments that the authors should clarify.

First, I feel that the phrase, "social motivations" do not reflect the meaning that how people are motivated to identify articles that would be liked by members of their political party if they shared them. I think the terminology should contain the meaning of "partisan -biased." Also, in the Abstract and the place in which the term "social motivations" first appears, please clarify what this means.

The reasons why we should consider "mixed motivations" in this study were not explained in the Intro section. I understand that "mixed motivations" are different from "social" and "accuracy" motivations, but why we should see this motivation separately (beyond the simple explanations that mixed motivations are a combination of social and accuracy motivations)?

On p.3 line 55, the sentence "However, it is unclear why this partisan divide in belief exists"  This should be revised, given that this has been examined in much prior research (e.g., motivated reasoning), as the authors explained on the following sentence.

The authors should come up with more clear justifications on why accuracy/social/and mixed motivations should be taken into account to reduce the impact of misinformation.

Clarify whether Experiment 1 and 2 also allowed U.S. adults to participate in this study only. Those are not nationally representative sample, right?

On p.8, line 238: Clarify why there are four conditions.

One major issue with this manuscript is feasibility of adopting these motivations for people to discern true/false news headlines. It seems to be impossible to motivate people every time they are exposed to news headlines. Can the authors suggest realistic ways on how we can utilize your findings for our real media environments?

The authors should be more specific on adding implications on non-significant interactions as well (e.g., accuracy incentives x source cues).

Overall, I think the findings of this research that "motivations" to be accurate are something that important to reduce the impact of misinformation, especially to conservatives.

Reviewer #2:

Remarks to the Author:

This manuscript is another iteration of analyses that have been published by different teams of researchers over the past few years. The idea of incentivizing cognitive effort has longstanding roots in psychology (e.g., work on accuracy nudges by David Rand), economics (e.g., Colin Camerer's work on incentives) and behavioral economics (most recently, work by people like Katie Milkman and others).

This paper offers yet another minor variation on this theme with many of the shortcomings that have characterized countless post-Trump attempts to counter misinformation. Let me address each of them in greater detail.

First, the authors set up their paper with the kind of panic around misinformation that is currently in vogue among many social scientists. The paper opens with an overview of the alleged effects that being misinformed might have on attitudes like vaccine hesitancy. Interestingly, the source for this claim is a study in *Nature Human Behavior* that does not show that being misinformed increases vaccine hesitancy. It showed that if academic researchers force participants in experiments to be exposed to misinformation that the researchers claim is true (e.g., statements about vaccine safety), captive audiences that are being fed misinformation by researchers differ in their attitudes measured in the subsequent questions from participants that are not being fed misinformation. Not surprisingly, this finding is inconsistent with other work (e.g., <https://www.nature.com/articles/s41541-021-00335-2>) that "found no evidence that belief in misinformation about COVID-19 treatments was positively associated with vaccine hesitancy."

Long story short, the somewhat hyperbolic introduction needs a much more nuanced overview of the literature and of the conclusions we can and cannot draw for real-world phenomena. This is also the case for the authors' somewhat sweeping statement about conservatives being more susceptible than liberals to misinformation. Every single study they cite to support this claim has been conducted during time periods and for issues (e.g., vaccines or COVID-19, more broadly) for which conservative views were heavily tainted by a Donald Trump campaign or administration. This time period is also one during which Democrats have signaled very explicitly that science is part of their tribal identity (march for science, Biden's "science is back" statement, etc.). Meanwhile, Trump has signaled equally strongly that following his lead on misguided COVID-19 claims is part of being non-RINO Republican.

In other words, Republican voters are likely not more susceptible to misinformation, they might just have follow current party leadership that endorses these claims. The claim by the authors (allegedly backed by the Imhoff et al. study) that similar partisan gaps show up in other countries is very misleading. Imhoff et al. deal with conspiracy beliefs, which is an entirely different beast from misinformation.

Second, this leaves us with yet another iteration of lab-experimental work that claims to test real-world solutions to an alleged misinformation problem. In this case, similar analyses and claims have been presented in a number of studies before. Since this study is not set up as a direct replication, this leaves us with the question of what's new. And the answer is: very little. The findings are somewhat consistent with internal validity-focused work examining human decision making in highly artificial and likely-short term settings. That's it.

And multiple iterations of research can have value, especially if they help us disentangle unresolved questions from previous work. Unfortunately, even the internal-validity focused claims of this paper are weak at best. As the authors acknowledge in the supplementary materials, there are a number of alternative interpretations that the paper wasn't able to test, given the constraints of the design, ranging from partisans giving what they know to be incorrect or correct answers, even if they contradict their personal beliefs, simply based on what they think they are expected to say. The authors offer a very naïve non-solution by asking respondents explicitly if they engaged in this behavior and conclude that this is not a concern since "[o]nly 5% of participants admitted to engaging in this kind of motivated responding, indicating that most participants reported answering in line with their genuine beliefs." I strongly disagree. One in 20 people in their sample were willing to say that they lie outright. How many people engaged in the same behavior but didn't admit to it? In fact, the finding that conservatives responded more strongly to incentives suggests to me that they know that the views they might endorse are not 100 percent factually correct. They hold them as an expression of partisan identity, not as a reflection of their understanding. The U.S. General Social Survey, for example, has shown very different patterns of endorsement among the public for statements about evolution being true or being true "according to scientists." The much higher public endorsement for the "according to scientists" version shows nicely that people know what they're "supposed" to know, but refuse to say it in most cases. This is likely the same confound that the authors of this study are picking up on. I would strongly suggest that the authors do a deeper dive into the social desirability literature and especially George Bishop's work on "The Illusion of Public Opinion: Fact and Artifact in American Public Opinion Polls."

The applicability of highly artificial lab experiments also raises questions about how useful these findings are for real world interventions. How do we know that highly artificial lab experiments with captive audiences scale to the real world? The same controls that allow researchers to maximize internal validity in experiments also reduce their ecological validity. How do we know if an incentive offered by the Biden government (especially with significant partisan trust differentials for the current CDC) would help overcome vaccine hesitancy with conservatives in 2022? Would it not be reasonable to assume that such nudges or incentives would be ridiculed or decried as government overreach by conservative talk radio and TV commentators, producing either a null effect or even a backlash, with CDC losing even more credibility? This complete absence of any awareness among the authors of sociopolitical dimensions of the problem is surprising.

In short, what we have is yet another study tackling a well-studied problem in a highly artificial setting. The study does little to disentangle the same problems that previous in this space has

encountered and repeats some of the same mistakes.

Reviewer #3:

Remarks to the Author:

This manuscript reports three studies that examined the effects of financial incentives for accuracy on participants' accuracy and partisan bias in judgments of true and false headlines. In Experiment 1, participants in the accuracy incentive condition were more accurate and less biased than those in the control condition. Experiment 2 also included a social incentive condition and a mixed incentive condition. The main findings from Experiment 1 were replicated. Experiment 3 added the variable of source cues, again replicating the main results of Experiments 1 and 2. In all three experiments, the effect of accuracy incentives on discernment was driven by politically incongruent true headlines. An integrative analysis showed that the incentive reduced the difference in discernment between Republicans and Democrats.

I enjoyed reading this paper. The general topic of susceptibility to fake news is an important one, and the specific aim of understanding factors that contribute to differences in this susceptibility among liberals and conservatives, is understudied. This manuscript makes a novel and important contribution to this area.

The limitations of this study were adequately addressed in the Discussion: alternative explanations, specific stimuli used, and generalizability outside of the United States. Other concerns I had while reading the manuscript were addressed in the Supplementary Materials.

One clarification I have is about the specific dependent variable for the accuracy ratings. Participants rated each headline on a 1-6 scale. Line 132 cites a SDT paper (Batailler et al., 2021). I assume this meant that the authors calculated d' (or some other SDT measure of sensitivity)? The condition means are around 3.0, so I'm guessing this isn't d' . These means are likely difference scores (between true and false headlines), but some clarification would help. Also, the y-axis of Figure 1 is "percent rated as accurate." How was this calculated from Likert scale responses?

This minor clarification is all that I have. I found this article to be exceptionally clear and well-written.

Dustin Calvillo

Reviewer #4:

Remarks to the Author:

Thank you for the opportunity to review "Accuracy and Social Motivations Shape Judgements of (Mis)Information" for NHB. The paper offers evidence from 3 online experiments manipulating accuracy and social motives with small monetary incentives. It shows that accuracy incentives improve truth discernment and reduce partisan bias, particularly among conservatives, and specifically by increasing beliefs in politically incongruent true news. The paper also reports a number of interesting follow-up analyses on the psychological correlates of discernment and bias. Overall, the

findings underpin the importance of (partisan and social) motivations. There is much to like about this paper, it addresses an important and ongoing debate in the literature; its experiments are well-designed; the analyses are rigorous (although see issues about reporting). Yet, I also see some room for improvement, most of which is doable with some editing.

1. I very much agree with the framing of the paper that there is conflicting evidence on the role of various motivations in sharing and believing misinformation. line 85 calls for a "systematic investigation". line 337-8 argue that "these results stand in contrast to accounts of fake news belief that downplay the roles of motivation and partisanship". I applaud this intention, yet I wonder if more could be done to explain the conflicting results between previous research (and particularly, the Pennycook and Rand oeuvre) and the present findings. I think that if the paper could offer a convincing argument for why it should be better trusted than previous efforts to study these questions, if it went the extra mile to engage with the literature a bit more, it would solidify the contribution of the paper to earn a place at a prestigious outlet as NHB. Put differently (and perhaps too flippantly), Pennycook and Rand built their view that partisan motivation does not matter on dozens of experiments. Does this paper "merely" present 3 experiments which say otherwise? Or are there reasons to trust them more than prior work?

2. On experimental design, I wonder if the weak/mixed results of accuracy incentives on sharing intentions could be potentially explained by the fact that participants were asked to think about veracity beforehand. Pennycook and Rand show that even a single accuracy nudge improves sharing intentions. Perhaps more importantly, I worry about the power for the novel hypothesis in Experiment 3, concerning the interaction between source cues and incentive manipulation (H4 in the prereg). Neither the pre-registration, nor the Methods section say much about the power calculation. Yet, I think Gelman's (in)famous maxim about 16x the sample size for interactions works well here. In a nutshell, interactions are by default twice as noisy as main effects ($SE = 4SD/\sqrt{N}$ instead of $2SD/\sqrt{N}$). To make things worse, the effect we can realistically expect is probably smaller than for main effects: an attenuation is more likely than a complete elimination of the effect. If for convenience we expect a 50% attenuation, we are after half the effect. So we need a sample to get 4x more precise estimates. Given that annoying square root over N, it means 16x the sample size. If the main experiments warranted $2 \times 216 = 432$ participants, we would need $432 \times 16 = 6912$ here. But the actual decrease in the effect is larger, around 66%. This still warrants 3x more precise estimates, and 9x larger sample (at around 4K). That said, this is just a quick, back of the envelope calculation, perhaps it's off. If not, however, then I would treat all interactions reported in the paper, even those on the pooled sample with a certain caution. Is it true that CRT (knowledge etc) does not moderate the effect of the intervention, or is it only that it lacks power to detect realistic effects? As a post-hoc sensitivity test, you can multiply the SE estimates from OLS regressions with 2.8 to get a sense of an effect size which could be detected with 80% power (see Howard S. Bloom. Minimum Detectable Effects: A Simple Way to Report the Statistical Power of Experimental Designs. Evaluation Review).

Gelman's blog on power and interactions: <https://statmodeling.stat.columbia.edu/2018/03/15/need-16-times-sample-size-estimate-interaction-estimate-main-effect/> A more academic treatment can be found in his (et al) textbook, Regression and Other Stories. Blake & Gangestad (2020) PSPB uses a different logic to arrive at similar conclusions.

3. On reporting, I would encourage more precision in language, and particularly to avoid dichotomizing effects based on 5% alpha cutoff. Just because we cannot reject the null for a given test, it does not mean that there is no difference between two conditions. e.g. on lines 248-9 "truth

discernment was not higher in the accuracy incentives condition ($M = 2.15, \dots$) compared to the control condition ($M = 2.00$). But of course, $2.15 > 2.00$. "truth discernment was not *significantly* higher ... " would be better, but whether that's what we substantively care about is a separate issue. For statistical evidence for no difference, equivalence tests could be used (Lakens 2017, SPSS). I would also avoid using "marginal" for $p > 0.05$, which is a red flag for some readers. The authors should also remember the maxim that "The Difference Between "Significant" and "Not Significant" is not Itself Statistically Significant" (Gelman and Stern 2006). Thus, I think the right conclusion from Experiment 3 is that omitting source cues appear to slightly reduce the effect of accuracy incentives, yet the experiment is inconclusive as to whether this is due to chance or not. I am not sure the conclusion in line 349 is well supported by the data.

4. I think Figs 1 and 2 could be improved. Although I welcome reporting and visualizing relatively raw data, perhaps it could be relegated to the online appendix here (along with similar plots for sharing intentions), and figures which communicate the main findings better could be drawn for the main text. My main qualm is that truth discernment is the difference between the blue and yellow bars, which is hard to eyeball, let alone compare across conditions. Partisan bias is even more tricky. Reporting these (discernment + bias) directly by condition would help a lot. Also, I think it would be much more intuitive to show control first (on the left) and then the treatment group. I repeatedly had to fight the (first) impression that accuracy decreases from accuracy group to control. I also wonder why Table 1 is not a plot. I think it should be.

Finally, a few smaller issues:

l108 "sources cues" typo?

l144 concerns politically-incongruent *true* headlines, right?

p7 I missed a direct comparison between social incentive versus control, and a discussion of the results.

l239 I would reformulate the sentence or at least delete the parentheses to clarify

l380 perhaps reformulate to acknowledge that the paper is about sharing *intentions* not sharing per se.

Methods. I would have preferred to read at least a few sample headlines.

l553 & 555 talk about (social vs control), yet Exp 3 had not manipulated this, right?

Alexander Bor

Author Rebuttal to Initial comments

REVIEWER COMMENTS

Reviewer #1:

Remarks to the Author:

The authors are to be commended for choosing a topic of such high importance and social relevance and collecting a large sample size for three experiments. I'm generally satisfied with the authors' approach, and I think the research findings provide significant implications to the current misinformation literature. Below I have some comments that the authors should clarify.

Thank you for these comments. We are glad that you believe the findings have significant implications for the current misinformation literature.

First, I feel that the phrase, "social motivations" do not reflect the meaning that how people are motivated to identify articles that would be liked by members of their political party if they shared them. I think the terminology should contain the meaning of "partisan -biased." Also, in the Abstract and the place in which the term "social motivations" first appears, please clarify what this means.

We agree that the term “social motivations” was too broad. Following your recommendation, we now refer to that condition as the “partisan-identity” motivation condition to more precisely describe what this condition is experimentally manipulating. On pg. 7, we write:

“In addition to the *accuracy incentives* and *control condition*, we added a *partisan identity motivation* condition, whereby participants were given a financial incentive to correctly identify articles that would appeal to members of their own political party.”

Instead of saying “social motivations” in the abstract, we now say:

“Incentivizing people to identify news that would be liked by their political allies, however, decreased accuracy.”

The reasons why we should consider "mixed motivations" in this study were not explained in the Intro section. I understand that "mixed motivations" are different from "social" and "accuracy" motivations, but why we should see this motivation separately (beyond the simple explanations that mixed motivations are a combination of social and accuracy motivations)?

Thank you for the suggestion to make this clearer. We have now included more explanation about the motivation behind this condition. The Identity-Based Model of Political Belief (Van Bavel & Pereira, 2018) proposes that partisan-identity goals can sometimes outweigh accuracy goals, which may lead people to hold inaccurate (but partisan-congruent) beliefs. Building off of the predictions of this model, we wanted to test whether making partisan-identity goals salient, while also making accuracy salient, decreased the effect of the accuracy incentives. In other words, we wanted to see whether partisan-identity goals interfered with accuracy goals in this condition. We write more about this on pg. 7:

“Building off of the predictions of the identity-based model of political belief¹³, we wanted to examine whether increasing social or partisan-identity related goals might interfere with accuracy goals. Thus, in a final condition, called the *mixed motivation condition*, participants received a financial incentive of up to one dollar to identify articles that would be liked by one’s in-group, followed by an additional financial incentive to identify accurate articles.”

On p.3 line 55, the sentence "However, it is unclear why this partisan divide in belief exists"  This should be revised, given that this has been examined in much prior research (e.g., motivated reasoning), as the authors explained on the following sentence.

Thank you for this suggestion. We have now revised this sentence to say on pg. 3: “However, there are multiple possible reasons that can explain why this partisan divide in belief exists.”

The authors should come up with more clear justifications on why accuracy/social/and mixed motivations should be taken into account to reduce the impact of misinformation.

We have now added a clearer discussion about the practical implications of our research. While much of this work tests theoretical questions, such as the extent to which motivation versus prior knowledge influence belief in true and false news, it also has a number of important practical implications, demonstrating that motivation-based interventions (as opposed to, or in addition to, media-literacy interventions) may be able to decrease the belief and sharing of fake news. We discuss in more depth on *pg. 14*:

“These results also have practical implications for interventions^{73,74}. Accuracy incentives improved the accuracy of people’s judgements, and an integrative data analysis found that this effect may have spilled over into intentions to share more accurate articles. However, the effect on sharing intentions was small

and inconsistent across studies. This may be in part because people were asked about accuracy before being asked about sharing intentions, and past research has found that merely asking people about accuracy can improve the accuracy of sharing intentions⁶⁵. Further, making partisan-identity motivations salient increased the sharing of both politically-congruent false (and true) news. Thus, interventions and social media design features should aim to both *increase* accuracy motivations and *decrease* motivations to share inaccurate content that receives high social reward.”

Clarify whether Experiment 1 and 2 also allowed U.S. adults to participate in this study only. Those are not nationally representative sample, right?

We have clarified that Experiments 1 and 2 used politically-balanced US samples of adults that are not nationally representative. Experiment 3 uses a nationally-representative sample, and Experiment 4 also uses a politically-balanced US sample of adults.

On pg. 4, we write: “In Experiment 1, we recruited a politically-balanced sample of 462 US adults via the survey platform Prolific Academic.”

On pg. 7, we write: “We recruited another politically-balanced sample of 998 US adults.”

On pg. 8, we write: “In Experiment 3, we aimed to replicate our prior findings in a nationally representative sample in the United States and test a potential process behind the effects of accuracy incentives. Specifically, we recruited a sample of 921 participants that was quota-matched to the national distribution on age, gender, ethnicity, and political party.”

On pg. 8, we also write: “In Experiment 4 we replicated the accuracy incentive and control condition in another politically-balanced sample of 983 US adults.”

Additionally, the methods section provides greater detail about the make-up of our samples.

On p.8, line 238: Clarify why there are four conditions.

We have clarified that there are four conditions because we are testing the impact of accuracy motivations (versus a control condition) with and without source cues on the headlines. We now say on pg. 8:

“Because we wanted to compare the effects of accuracy incentives with and without sources, this study had four conditions: accuracy incentives (with sources), control (with sources), accuracy incentives (without sources), and control (with sources).”

One major issue with this manuscript is feasibility of adopting these motivations for people to discern true/false news headlines. It seems to be impossible to motivate people every time they are exposed to news headlines. Can the authors suggest realistic ways on how we can utilize your findings for our real media environments?

This is an excellent point, which we considered carefully when designing our fourth experiment. Whereas it is unfeasible to pay people to be accurate every time there is a false news headline, it is more feasible to motivate people to be accurate in other ways. For example, one could emphasize that accuracy is a social norm, tell people that being inaccurate can harm their reputation, or give people feedback on how accurate they are. Certain design elements of social media can be changed to make accuracy motivations more relevant than social motivations. Experiment 4 suggests that other approaches to motivating accuracy beyond providing financial incentives may be beneficial. We discuss the practical implications of our work on page 9:

“Together, these results suggest that a subtler (and also more scalable) accuracy motivation intervention that does not employ financial incentives may be effective at increasing the perceived accuracy of true news from the opposing party, but appears to have a smaller effect size than the stronger financial incentive intervention.”

We also discuss the practical implications in detail in the discussion section, on pg. 14:

“These results also have practical implications for interventions^{73,74}. Accuracy incentives improved the accuracy of people’s judgements, and an integrative data analysis found that this effect may have spilled over into intentions to share more accurate articles. However, the effect on sharing intentions was small and inconsistent across studies. This may be in part because people were asked about accuracy before being asked about sharing intentions, and past research has found that merely asking people about accuracy can improve the accuracy of sharing intentions⁶⁵. Further, making partisan-identity motivations salient increased the sharing of both politically-congruent false (and true) news. Thus, interventions and social media design features should aim to both *increase* accuracy motivations and *decrease* motivations to share inaccurate content that receives high social reward. While effects were only found for false (and not true) headlines, people tend to encounter blatantly false news very infrequently⁶⁸, leading some to suggest that increasing trust in reliable news is more important than reducing belief in falsehoods⁷⁵ and that researchers should employ a broad definition of misinformation⁷⁶.”

The authors should be more specific on adding implications on non-significant interactions as well (e.g., accuracy incentives x source cues).

We agree, and have added much more detail about implications for non-significant interactions. For example, we say on pg. 8:

“Although effect sizes appeared to be descriptively smaller when sources were removed from the headlines (see **Fig. 1** and *Supplementary Appendix S1* for detail), we did not find significant interactions between the main outcome variables and the presence or absence of source cues. However, this study design did not provide strong power to test whether this was not due to chance, since interaction effects can require up to 16 times as much power as main effects^{57,58} (see *Methods* for power analysis). Additional analysis using Bayes factors⁵⁹ reported in *Supplementary Appendix S1* did not find strong evidence for the absence of interaction effects.”

We also include a power analysis for interaction effects in the Methods section on pg. 19: “Based on the effect sizes of Study 2 and the principle that 16 times the sample size is needed to detect an attenuated interaction effect^{74,75}, a post-hoc power analyses conducted after we ran the study found that we needed roughly 1536 participants to detect an interaction for the amount of politically-incongruent news rated as true, 2560 participants to detect an interaction effect for truth discernment, and 7488 participants to detect an interaction effect for partisan bias with 80% power. Thus, this particular design was underpowered to detect whether accuracy incentives interacted with source cues.”

We also added additional analysis using Bayes Factors in the supplementary appendix (pg. 4):

“We then conducted Bayesian analyses for the null interactions using the “BayesFactor” package in R. This analysis used a noninformative Jeffreys prior on the variance of the normal population, and a Cauchy prior on the standardized effect size. The Bayes factor for the interaction between accuracy incentives and source cues on the amount of incongruent true news rated as true was 0.27 ($\pm 4.37\%$), the Bayes factor for this interaction on partisan bias was 0.10 ($\pm 3.76\%$), and the Bayes factor for this interaction on truth discernment was 0.18 ($\pm 5.51\%$). Bayes factors in this range indicate weak (or “anecdotal”) to moderate levels of evidence for the null hypothesis. Thus, this study does not provide strong evidence for null interaction effects, but also does not provide strong evidence for moderation effects.”

Overall, I think the findings of this research that "motivations" to be accurate are something that

important to reduce the impact of misinformation, especially to conservatives.

We are glad that you find this research to be important, and we thank you for the excellent and thoughtful feedback.

Reviewer #2:

Remarks to the Author:

This manuscript is another iteration of analyses that have been published by different teams of researchers over the past few years. The idea of incentivizing cognitive effort has longstanding roots in psychology (e.g., work on accuracy nudges by David Rand), economics (e.g., Colin Camerer’s work on incentives) and behavioral economics (most recently, work by people like Katie Milkman and others). This paper offers yet another minor variation on this theme with many of the shortcomings that have characterized countless post-Trump attempts to counter misinformation. Let me address each of them in greater detail. First, the authors set up their paper with the kind of panic around misinformation that is currently in vogue among many social scientists. The paper opens with an overview of the alleged effects that being misinformed might have on attitudes like vaccine hesitancy. Interestingly, the source for this claim is a study in *Nature Human Behavior* that does not show that being misinformed increases vaccine hesitancy. It showed that if academic researchers force participants in experiments to be exposed to misinformation that the researchers claim is true (e.g., statements about vaccine safety), captive audiences that are being fed misinformation by researchers differ in their attitudes measured in the subsequent questions from participants that are not being fed misinformation. Not surprisingly, this finding is inconsistent with other work (e.g., <https://www.nature.com/articles/s41541-021-00335-2>) that “found no evidence that belief in misinformation about COVID-19 treatments was positively associated with vaccine hesitancy.”

Thank you for these comments. We have toned down some of the language in the introduction, and cite your recommended suggested source to add some nuance. While we appreciate suggestions to add nuance and acknowledge null effects, we believe it is well-established that repeated exposure to misinformation tends to increase belief, and have included more citations to support this. We also add some sources suggesting belief in misinformation is widespread to help motivate our investigation. See our updated introduction on page 3:

“Misinformation – which can refer to fabricated news stories, false rumors, conspiracy theories, or disinformation campaigns – can have serious negative effects on society and democracy^{1,2}. Numerous

studies suggest that misinformation exposure may reduce support for climate change^{5,6} and the COVID-19 vaccine^{7,8}, and that the mere repetition of misinformation can increase belief in it^{9,10}. Anti-vaccination viewpoints are becoming increasingly popular online¹¹, and there is widespread belief in misinformation and conspiracy theories about election fraud¹² and COVID-19¹³. There has thus been a growing interest in understanding the psychology of belief in misinformation and how to mitigate its spread^{1,2,14–16}.”

We cite the null result you suggested in a footnote: “It should be noted that there are some null results regarding the effects of misinformation, such as a correlational study finding that belief in COVID-19 misinformation was not associated with vaccine hesitancy³ In contrast, a recent study in 69 countries found that false beliefs about COVID-19 were a robust predictor of behavioral intentions towards public health behavior.” While null results are to be expected (and should be cited), we do not believe this null result contradicts the general claim that misinformation exposure can influence belief or behavior. Indeed, study you mentioned was correlational study with a sample of 1000 people, but the other correlational study we cited showed in a cross-cultural sample of over 50,000 people that misinformation belief correlates with preventative health behaviors. We have also cited multiple meta-analyses in the introduction show that address the claim that misinformation exposure can affect beliefs and behaviors. Nonetheless, we hedge our language to acknowledge uncertainty and try to present the literature in a more neutral manner.

Long story short, the somewhat hyperbolic introduction needs a much more nuanced overview of the literature and of the conclusions we can and cannot draw for real-world phenomena. This is also the case for the authors’ somewhat sweeping statement about conservatives being more susceptible than liberals to misinformation. Every single study they cite to support this claim has been conducted during time periods and for issues (e.g., vaccines or COVID-19, more broadly) for which conservative views were heavily tainted by a Donald Trump campaign or administration. This time period is also one during which Democrats have signaled very explicitly that science is part of their tribal identity (march for science, Biden’s “science is back” statement, etc.). Meanwhile, Trump has signaled equally strongly that following his lead on misguided COVID-19 claims is part of being non-RINO Republican. In other words, Republican voters are likely not more susceptible to misinformation, they might just have follow current party leadership that endorses these claims. The claim by the authors (allegedly backed by the Imhoff et al. study) that similar partisan gaps show up in other countries is very misleading. Imhoff et al. deal with conspiracy beliefs, which is an entirely different beast from misinformation.

Thank you for these thoughtful points. We believe our updated introduction now includes more nuance. We have clarified that these studies are primarily about US conservatives. We have also added your thoughtful point that we do not know whether partisan asymmetries reflect the current political climate or cross-cultural trends in thinking style on page 14:

“Future work could examine whether this asymmetry arises due to the dynamics of partisan identity, party leadership, and social norms in the United States during this specific political climate, or if it reflects broader differences between liberals and conservatives that can be observed across cultures.^{71,72}”

We also removed the reference to the Imhoff et. al (2022) study from the introduction and moved it to the sentence above in the discussion. We agree that the Imhoff study does not speak to misinformation belief in particular and is more broadly about conspiracy theories, which is related to misinformation belief but conceptually different. Thank you for your suggestion to be more precise about these issues.

Second, this leaves us with yet another iteration of lab-experimental work that claims to test real-world solutions to an alleged misinformation problem. In this case, similar analyses and claims have been presented in a number of studies before. Since this study is not set up as a direct replication, this leaves us with the question of what’s new. And the answer is: very little. The findings are somewhat consistent with internal validity-focused work examining human decision making in highly artificial and likely-short term settings. That’s it.

Our research is highly novel and has key differences to other bodies of work. But, we do agree that we should have made our novel contribution clearer. In this revision, we have aimed to more clearly articulate two key theoretical contributions and one practical contribution in the below paragraphs (pg. 14):

“These results make two key theoretical contributions. First, they suggest that partisan differences in news judgements do not simply reflect differences in factual knowledge¹⁴. Instead, our data suggest that a substantial portion of this partisan divide can be attributed to a lack of motivation to be accurate. While there have been debates about whether partisan differences in belief reflect differing prior beliefs versus politically-motivated cognition,^{25,26} our studies provide causal evidence for the effect of motivation on belief. Along with other research^{41,69,70}, these findings suggest that survey data about belief in (mis)information should not be taken at face value, because people answer survey questions differently when they are highly motivated to be accurate. However, judgements of false headlines appeared to be unaffected by accuracy motivations, suggesting that other factors may play a more prominent role in people’s assessment of false news as compared to true news.

Second, while a number of studies have observed that American conservatives tend to be more susceptible to misinformation than liberals²⁹⁻³⁴, our studies find that the gap in accuracy between liberals and (unincentivized) conservatives closes by more than half when conservatives are motivated to be accurate. Future work could examine whether this asymmetry arises due to the dynamics of partisan

identity, party leadership, and social norms in the United States during this specific political climate, or if it reflects broader differences between liberals and conservatives that can be observed across cultures^{71,72}.

These results also have practical implications for interventions^{73,74}. Accuracy incentives improved the accuracy of people’s judgements, and an integrative data analysis found that this effect may have spilled over into intentions to share more accurate articles. However, the effect on sharing intentions was small and inconsistent across studies. This may be in part because people were asked about accuracy before being asked about sharing intentions, and past research has found that merely asking people about accuracy can improve the accuracy of sharing intentions⁶⁵. Further, making partisan-identity motivations salient increased the sharing of both politically-congruent false (and true) news. Thus, interventions and social media design features should aim to both *increase* accuracy motivations and *decrease* motivations to share inaccurate content that receives high social reward. While effects were only found for false (and not true) headlines, people tend to encounter blatantly false news very infrequently⁶⁸, leading some to suggest that increasing trust in reliable news is more important than reducing belief in falsehoods⁷⁵ and that researchers should employ a broad definition of misinformation⁷⁶.”

We have also brought up your important points about external validity and addressed them in depth. On pg. 14, we write:

“One limitation of this work is that survey experiments have unknown ecological validity. To maximize ecological validity, we used real, pre-tested news headlines in the format in which they would be regularly encountered on social media websites such as Facebook. Additionally, self-reported sharing intentions are highly correlated with real online news sharing⁷⁷, and a field experiments suggests that priming accuracy can improve news sharing decisions on Twitter⁶⁵, illustrating that results from survey experiments on misinformation can translate to the field.”

And multiple iterations of research can have value, especially if they help us disentangle unresolved questions from previous work. Unfortunately, even the internal-validity focused claims of this paper are weak at best. As the authors acknowledge in the supplementary materials, there are a number of alternative interpretations that the paper wasn’t able to test, given the constraints of the design, ranging from partisans giving what they know to be incorrect or correct answers, even if they contradict their personal beliefs, simply based on what they think they are expected to say. The authors offer a very naïve non-solution by asking respondents explicitly if they engaged in this behavior and conclude that this is not a concern since “[o]nly 5% of participants admitted to engaging in this kind of motivated responding, indicating that most participants reported answering in line with their genuine beliefs.” I strongly disagree. One in 20 people in their sample were willing to say that they lie outright. How many people engaged in the same behavior but didn’t admit to it? In fact, the finding that conservatives responded more strongly to incentives suggests to me that they know that the views they might endorse are not 100 percent factually

correct. They hold them as an expression of partisan identity, not as a reflection of their understanding. The U.S. General Social Survey, for example, has shown very different patterns of endorsement among the public for statements about evolution being true or being true “according to scientists.” The much higher public endorsement for the “according to scientists’ version shows nicely that people know what they’re “supposed” to know, but refuse to say it in most cases. This is likely the same confound that the authors of this study are picking up on. I would strongly suggest that the authors do a deeper dive into the social desirability literature and especially George Bishop’s work on “The Illusion of Public Opinion: Fact and Artifact in American Public Opinion Polls.”

Thank you for your thoughtful point about alternate explanations. We have considered it thoroughly and have decided to add another study that helps address alternate explanations. Experiment 4 finds that increasing people’s motivation to be accurate in non-financial ways is also effective at increasing the perceived accuracy of politically-incongruent true news. This helps rule out the explanation that our effects are completely driven by people stating that certain headlines are true simply to get money, since people were not compelled to respond in any particular way in this condition. We address alternate explanations on page 14:

“Another potential limitation is that there are multiple ways to interpret the effects of financial incentives. For instance, people may be guessing what they think fact-checkers believe in order to get money, rather than expressing their true beliefs. However, this interpretation is unlikely to explain the full effect, since a subtle non-financial accuracy motivation intervention had similar (albeit slightly smaller) effects. Furthermore, supplementary analysis found that an extremely small percentage of participants reported answering in ways that did not accord with their true beliefs to get money (See *Supplementary Appendix SI*).”

We have looked in depth at the social desirability literature and have added the source that you suggested for us on page 14:

“Along with other research^{41,69,70}, these findings suggest that survey data about belief in (mis)information should not be taken at face value, because people answer survey questions differently when they are highly motivated to be accurate.”

The applicability of highly artificial lab experiments also raises questions about how useful these findings are for real world interventions. How do we know that highly artificial lab experiments with captive audiences scale to the real world? The same controls that allow researchers to maximize internal validity in experiments also reduce their ecological validity. How do we know if an incentive offered by the Biden government (especially with significant partisan trust differentials

for the current CDC) would help overcome vaccine hesitancy with conservatives in 2022? Would it not be reasonable to assume that such nudges or incentives would be ridiculed or decried as government overreach by conservative talk radio and TV commentators, producing either a null effect or even a backlash, with CDC losing even more credibility? This complete absence of any awareness among the authors of sociopolitical dimensions of the problem is surprising.

In short, what we have is yet another study tackling a well-studied problem in a highly artificial setting. The study does little to disentangle the same problems that previous in this space has encountered and repeats some of the same mistakes.

We do not recommend actually paying people to be accurate in the real world, and we agree this approach could backfire or simply be difficult to scale. The financial incentive paradigm mostly tells us how people would answer if they were highly motivated to be accurate, which allows us to test important theoretical questions. But, we believe that aspects of social media platform design can be improved to subtly encourage accuracy motivations and decrease motivations to share inaccurate content. Study 4 addresses scalable ways to motivate accuracy in the real world.

We have considered conducting field experiments related to this line of work, though we were more interested in perceived accuracy judgements as opposed to sharing behavior, which cannot be measured in a field environment and have to be measured via surveys. We note that research on priming accuracy suggests that motivating people to be accurate can improve sharing behavior in the field, and we also note that sharing intentions in surveys correlate with real-world sharing behavior on page 14:

“Additionally, self-reported sharing intentions are highly correlated with real online news sharing,⁷² and a field experiment suggests that priming accuracy can improve news sharing decisions on Twitter,⁶² illustrating that results from survey experiments on misinformation can translate to the field.”

Reviewer #3:

Remarks to the Author:

This manuscript reports three studies that examined the effects of financial incentives for accuracy on participants' accuracy and partisan bias in judgments of true and false headlines. In Experiment 1, participants in the accuracy incentive condition were more accurate and less biased than those in the control condition. Experiment 2 also included a social incentive condition and a mixed incentive condition. The main findings from Experiment 1 were replicated. Experiment 3 added the variable of source cues, again replicating the main results of Experiments 1 and 2. In all three experiments,

the effect of accuracy incentives on discernment was driven by politically incongruent true headlines. An integrative analysis showed that the incentive reduced the difference in discernment between Republicans and Democrats.

I enjoyed reading this paper. The general topic of susceptibility to fake news is an important one, and the specific aim of understanding factors that contribute to differences in this susceptibility among liberals and conservatives, is understudied. This manuscript makes a novel and important contribution to this area.

Thank you very much. We are glad you think the paper makes a novel and important contribution.

The limitations of this study were adequately addressed in the Discussion: alternative explanations, specific stimuli used, and generalizability outside of the United States. Other concerns I had while reading the manuscript were addressed in the Supplementary Materials.

We are glad that you think the limitations were adequately discussed, and we also conducted another study to help rule out some alternate explanations.

One clarification I have is about the specific dependent variable for the accuracy ratings. Participants rated each headline on a 1-6 scale. Line 132 cites a SDT paper (Batailler et al., 2021). I assume this meant that the authors calculated d' (or some other SDT measure of sensitivity)? The condition means are around 3.0, so I'm guessing this isn't d' . These means are likely difference scores (between true and false headlines), but some clarification would help. Also, the y-axis of Figure 1 is "percent rated as accurate." How was this calculated from Likert scale responses?

We did not calculate d' and instead calculated difference scores (true headlines rated as true - false headlines rated as true), and partisan bias (politically-congruent headlines rated as true – politically-congruent headlines rated as false), which have been used in past studies (Pennycook & Rand, 2021). However, in this revision, we added signal detection analysis closely following Batailler et. al (2021) to the Supplementary Appendix as a robustness check, and found very similar results. We write on pg. 19 of the manuscript:

“As another robustness check, we also conducted supplemental analysis using signal detection modeling¹⁹. This analysis found that incentives increased people’s discrimination between true and false news (for both politically-congruent and politically incongruent headlines), and also increased the threshold by which people accepted politically-incongruent headlines as true (See *Supplementary*

Appendix S12). Thus, analysis using signal detection modeling yielded highly similar results to our main analysis.”

To calculate “percent rated as accurate,” we dichotomized accuracy ratings and calculated the percentage articles rated as true following Bago et al. (2020). The dichotomous analysis was pre-registered in experiments 2-4 since incentives were awarded only based on categorizing headlines as true or false, but supplementary analysis found that our key conclusions did not change when results were analyzed continuously (*Supplementary Appendix S8*). As such, the results appear robust to how this variable is operationalized. We describe how we calculate percentages in more depth in the Methods section (pg. 17):

“To calculate the percent of articles of a certain category rated as accurate, we divide the dichotomous accuracy scores by the total number of headlines from that category. For instance, if a person rated 50% of incongruent true articles as true, this means they responded “slightly accurate,” “moderately accurate,” or “extremely accurate” on two out of four incongruent true news articles.”

This minor clarification is all that I have. I found this article to be exceptionally clear and well-written.

Thank you, we are glad you find the paper to be clear and well-written.

Dustin Calvillo

Reviewer #4:

Remarks to the Author:

Thank you for the opportunity to review "Accuracy and Social Motivations Shape Judgements of (Mis)Information" for NHB. The paper offers evidence from 3 online experiments manipulating accuracy and social motives with small monetary incentives. It shows that accuracy incentives improve truth discernment and reduce partisan bias, particularly among conservatives, and specifically by increasing beliefs in politically incongruent true news. The paper also reports a number of interesting follow-up analyses on the psychological correlates of discernment and bias. Overall, the findings underpin the importance of (partisan and social) motivations. There is much to like about this paper, it addresses an important and ongoing debate in the literature; its experiments are well-designed; the analyses are rigorous (although see issues about reporting). Yet, I also see some room for improvement, most of which is doable with some editing.

Thank you for your kind words. We are glad you think the experiments are well-designed and that the reporting is rigorous.

1. I very much agree with the framing of the paper that there is conflicting evidence on the role of various motivations in sharing and believing misinformation. line 85 calls for a "systematic investigation". line 337-8 argue that "these results stand in contrast to accounts of fake news belief that downplay the roles of motivation and partisanship". I applaud this intention, yet I wonder if more could be done to explain the conflicting results between previous research (and particularly, the Pennycook and Rand oeuvre) and the present findings. I think that if the paper could offer a convincing argument for why it should be better trusted than previous efforts to study these questions, if it went the extra mile to engage with the literature a bit more, it would solidify the contribution of the paper to earn a place at a prestigious outlet as NHB. Put differently (and perhaps too flippantly), Pennycook and Rand built their view that partisan motivation does not matter on dozens of experiments. Does this paper "merely" present 3 experiments which say otherwise? Or are there reasons to trust them more than prior work?

In this revision we have engaged more with current literature on misinformation and more precisely present our contribution.

We have more clearly identified how our results contradict other accounts of news belief and sharing on pg. 13:

“While prominent accounts of misinformation sharing claim that partisanship and politically motivated cognition play a limited role in the belief and sharing of misinformation as compared to other factors (such as reflection or inattention)^{13,62}, our results, in contrast, indicate that motivation and partisan identity or ideology are indeed important factors. Our data point to the importance of broad theoretical accounts of (mis)information belief^{2,14,63} and sharing that integrate motivation and partisan identity with other variables^{2,13,14,27}. Indeed, an investigation using cognitive modeling found that a broad model of misinformation belief² that included multiple factors (such as partisan identity, cognitive reflection, and more) performed better at predicting acceptance of fake news than other models that included fewer variables⁶⁴.”

We have now significantly revised the discussion section to make the novel contribution of our work and its place in the literature clearer. We write on pg. 14:

“These results make two key theoretical contributions. First, they suggest that partisan differences in news judgements do not simply reflect differences in factual knowledge¹⁴. Instead, our data suggest that a substantial portion of this partisan divide can be attributed to a lack of motivation to be accurate. While there have been debates about whether partisan differences in belief reflect differing prior beliefs versus politically-motivated cognition,^{25,26} our studies provide causal evidence for the effect of motivation on belief. Along with other research^{41,69,70}, these findings suggest that survey data about belief in (mis)information should not be taken at face value, because people answer survey questions differently when they are highly motivated to be accurate. However, judgements of false headlines appeared to be unaffected by accuracy motivations, suggesting that other factors may play a more prominent role in people’s assessment of false news as compared to true news.

Second, while a number of studies have observed that American conservatives tend to be more susceptible to misinformation than liberals^{29–34}, our studies find that the gap in accuracy between liberals and (unincentivized) conservatives closes by more than half when conservatives are motivated to be accurate. Future work could examine whether this asymmetry arises due to the dynamics of partisan identity, party leadership, and social norms in the United States during this specific political climate, or if it reflects broader differences between liberals and conservatives that can be observed across cultures.^{71,72}

These results also have practical implications for interventions^{73,74}. Accuracy incentives improved the accuracy of people’s judgements, and an integrative data analysis found that this effect may have spilled over into intentions to share more accurate articles. However, the effect on sharing intentions was small and inconsistent across studies. This may be in part because people were asked about accuracy before being asked about sharing intentions, and past research has found that merely asking people about accuracy can improve the accuracy of sharing intentions⁶⁵. Further, making partisan-identity motivations salient increased the sharing of both politically-congruent false (and true) news. Thus, interventions and social media design features should aim to both *increase* accuracy motivations and *decrease* motivations to share inaccurate content that receives high social reward. While effects were only found for false (and not true) headlines, people tend to encounter blatantly false news very infrequently⁶⁸, leading some to suggest that increasing trust in reliable news is more important than reducing belief in falsehoods⁷⁵ and that researchers should employ a broad definition of misinformation⁷⁶.”

2. On experimental design, I wonder if the weak/mixed results of accuracy incentives on sharing intentions could be potentially explained by the fact that participants were asked to think about veracity beforehand. Pennycook and Rand show that even a single accuracy nudge improves sharing intentions.

This is an excellent point, and we agree. Since participants’ attention has already been drawn to accuracy, the extra incentive may not make a large difference.

On pg. 14, we write: “This may be in part because people were asked about accuracy before being asked about sharing intentions, and past research has found that merely asking people about accuracy can improve the accuracy of sharing intentions⁶²”

Perhaps more importantly, I worry about the power for the novel hypothesis in Experiment 3, concerning the interaction between source cues and incentive manipulation (H4 in the prereg). Neither the pre-registration, nor the Methods section say much about the power calculation. Yet, I think Gelman’s (in)famous maxim about 16x the sample size for interactions works well here. In a nutshell, interactions are by default twice as noisy as main effects ($SE = 4SD/\sqrt{N}$) instead of $2SD/\sqrt{N}$). To make things worse, the effect we can realistically expect is probably smaller than for main effects: an attenuation is more likely than a complete elimination of the effect. If for convenience we expect a 50% attenuation, we are after half the effect. So we need a sample to get 4x more precise estimates. Given that annoying square root over N, it means 16x the sample size. If the main experiments warranted $2 \times 216 = 432$ participants, we would need $432 \times 16 = 6912$ here. But the actual decrease in the effect is larger, around 66%. This still warrants 3x more precise estimates, and 9x larger sample (at around 4K). That said, this is just a quick, back of the envelope calculation, perhaps it’s off. If not, however, then I would treat all interactions reported in the paper, even those on the pooled sample with a certain caution. Is it true that CRT (knowledge etc) does not moderate the effect of the intervention, or is it only that it lacks power to detect realistic effects? As a post-hoc sensitivity test, you can multiply the SE estimates from OLS regressions with 2.8 to get a sense of an effect size which could be detected with 80% power (see Howard S. Bloom. Minimum Detectable Effects: A Simple Way to Report the Statistical Power of Experimental Designs. Evaluation Review). Gelman’s blog on power and interactions: <https://statmodeling.stat.columbia.edu/2018/03/15/need-16-times-sample-size-estimate-interaction-estimate-main-effect/> A more academic treatment can be found in his (et al) textbook, Regression and Other Stories. Blake & Gangestad (2020) PSPB uses a different logic to arrive at similar conclusions.

These are great points, and we make a number of changes to the manuscript to address them in depth. Specifically, as you note, we were underpowered to detect interaction effects in study 3 (according to the maxim that 16x the sample size is required to detect a main effect as compared to an interaction). We note that interactions between the accuracy incentives and source cues are largely inconclusive and that we were underpowered to detect interactions. We now include sample sizes as well as Bayes factors. See our discussion on pg. 8:

“Although effect sizes appeared to be descriptively smaller when sources were removed from the headlines (see Fig. 1 and *Supplementary Appendix S1* for detail), we did not find significant interactions

between the main outcome variables and the presence or absence of source cues. However, this study design did not provide strong power to test whether this was not due to chance, since interaction effects can require up to 16 times as much power as main effects^{57,58} (see *Methods* for power analysis). Additional analysis using Bayes factors⁵⁹ reported in *Supplementary Appendix S1* did not find strong evidence for the absence of interaction effects.”

We also note that, even with large sample sizes (of over 2000) in the integrative data analysis, we were still slightly underpowered to detect moderation effects. We now emphasize the moderation by ideology less strongly in this updated draft and encourage caution in interpreting this interaction. Also, supplementary analysis using Bayes factors did not find strong evidence for a moderation effect. See our discussion on pg. 12:

“However, even though we had a large sample, we were still slightly underpowered to detect these interaction effects (see power analysis in *Methods*), and supplemental Bayesian analyses also did not find strong evidence for the significant moderation effects (*Supplementary Appendix S11*), so they should be interpreted with caution.”

We also include a power analysis for the Study 3 interaction effects on pg. 18:

“Based on the effect sizes of Study 2 and the principle that 16 times the sample size is needed to detect an attenuated interaction effect^{74,75}, a post-hoc power analyses conducted after we ran the study found that we needed roughly 1536 participants to detect an interaction for the amount of politically-incongruent news rated as true, 2560 participants to detect an interaction effect for truth discernment, and 7488 participants to detect an interaction effect for partisan bias with 80% power. Thus, this particular design was underpowered to detect whether accuracy incentives interacted with source cues.”

We include another power analysis for the integrative data analysis interaction effects on pg. 19:

“Using effect sizes from the integrative data analysis and the principle that 16 times the sample size is needed to detect an attenuated interaction effect^{74,75}, a post-hoc power analysis found that we needed 2336 participants to detect an interaction effect for the amount of politically-incongruent news rated as true, 5984 participants to detect an interaction effect for truth discernment, 7488 for partisan bias, and 50,336 to detect an interaction for sharing discernment. Thus, moderation effects should be interpreted with caution.”

3. On reporting, I would encourage more precision in language, and particularly to avoid dichotomizing effects based on 5% alpha cutoff. Just because we cannot reject the null for a given test, it does not mean that there is no difference between two conditions. e.g. on lines 248-9 "truth discernment was not higher in the accuracy incentives condition ($M = 2.15$, ...) compared to the control condition ($M = 2.00$)". But of course, $2.15 > 2.00$. "truth discernment was not *significantly* higher ... " would be better, but whether that's what we substantively care about is a separate issue. For statistical evidence for no difference, equivalence tests could be used (Lakens 2017, SPPS). I would also avoid using "marginal" for $p > 0.05$, which is a red flag for some readers. The authors should also remember the maxim that "The Difference Between "Significant" and "Not Significant" is not Itself Statistically Significant" (Gelman and Stern 2006). Thus, I think the right conclusion from Experiment 3 is that omitting source cues appear to slightly reduce the effect of accuracy incentives, yet the experiment is inconclusive as to whether this is due to chance or not.

We agree that we should have been more precise in our language. Thank you very much for pointing this out. We avoid all use of the word "marginal," and we have adjusted all non-significant differences to specify that the interactions were not "significant." Also, following your suggestion, we update our conclusion about source cues on pg. 8:

"Although effect sizes appeared to be descriptively smaller when sources were removed from the headlines (see **Fig. 1** and *Supplementary Appendix S1* for detail), we did not find significant interactions between the main outcome variables and the presence or absence of source cues. However, this study design did not provide strong power to test whether this was not due to chance, since interaction effects can require up to 16 times as much power as main effects^{57,58}"

I am not sure the conclusion in line 349 is well supported by the data.

We agree that our claim was too strong and have removed it.

4. I think Figs 1 and 2 could be improved. Although I welcome reporting and visualizing relatively raw data, perhaps it could be relegated to the online appendix here (along with similar plots for sharing intentions), and figures which communicate the main findings better could be drawn for the main text. My main qualm is that truth discernment is the difference between the blue and yellow bars, which is hard to eyeball, let alone compare across conditions. Partisan bias is even more tricky. Reporting these (discernment + bias) directly by condition would help a lot. Also, I

think it would be much more intuitive to show control first (on the left) and then the treatment group. I repeatedly had to fight the (first) impression that accuracy decreases from accuracy group to control. I also wonder why Table 1 is not a plot. I think it should be.

We have redone the figures. Following your excellent suggestion, we now plot truth discernment and partisan bias, which we believe makes the findings clearer and simpler. We also follow your suggestion that the control condition should be shown on the left before the experimental condition. Since we do not emphasize the moderation by partisanship as much in this current draft because of power issues and lack of strong Bayes factors, and we have now moved Table 1 to the Supplementary Appendix (Table S5 on pg. 17 of the Supplementary Appendix). Thank you for all of these excellent suggestions.

Finally, a few smaller issues:

I108 "sources cues" typo?

We have fixed this typo.

I144 concerns politically-incongruent *true* headlines, right?

Yes, this has been fixed.

p7 I missed a direct comparison between social incentive versus control, and a discussion of the results.

We have added in this direct comparison, and a brief interpretation of the results on pg. 7:

“The partisan identity condition also did not significantly differ from the control condition ($p = 0.241$). Taken together, these results suggest that accuracy motivations increase truth discernment, but social motives can decrease truth discernment.”

I239 I would reformulate the sentence or at least delete the parentheses to clarify

We deleted the example and rephrased the sentence.

1380 perhaps reformulate to acknowledge that the paper is about sharing *intentions* not sharing per se.

We have changed all wording about sharing behavior to sharing intentions.

Methods. I would have preferred to read at least a few sample headlines.

We have added sample headlines to the methods section. On pg. 4, we write: “An example of a Democrat-leaning true headline was “Facebook removes Trump ads with symbols once used by Nazis” from *apnews.com*, and an example of a Democrat-leaning false news headline was “White House Chef Quits because Trump Has Only Eaten Fast Food For 6 Months” from *halfwaypost.com*.”

1553 & 555 talk about (social vs control), yet Exp 3 had not manipulated this, right?

Alexander Bor

You are correct – this was a typo, which we have now corrected. Thank you for pointing this out.

Decision Letter, first revision:

8th September 2022

Dear Mr Rathje,

Thank you once again for your revised manuscript, entitled "Accuracy and Social Motivations Shape Belief in (Mis)Information," and for your patience during the re-review process.

Your manuscript has now been evaluated by Reviewers 1, 3, and 4 from the original round of review. All reviewer feedback is included at the end of this letter. Although the reviewers found your manuscript to have improved during revision, they also raise some important outstanding concerns. We remain interested in the possibility of publishing your study in *Nature Human Behaviour*, but would

like to consider your response to these outstanding concerns in the form of a revised manuscript before we make a decision on publication.

Reviewers 1 and 3 are satisfied with the current revisions. However, Reviewer 4 raises a very important concern about your operationalization of partisan bias differing from earlier literature and this possibly affecting the results. Editorially and to rule out that the conclusions regarding the role of partisan motivations are driven by these differences in operationalization, we request that you provide a clear rationale for your operationalization and also re-analyse your data adhering to the alternative existing measure of partisan bias.

In sum, we invite you to revise your manuscript taking into account all reviewer and editor comments. We are committed to providing a fair and constructive peer-review process. Do not hesitate to contact us if there are specific requests from the reviewers that you believe are technically impossible or unlikely to yield a meaningful outcome.

We hope to receive your revised manuscript within 4-8 weeks. I would be grateful if you could contact us as soon as possible if you foresee difficulties with meeting this target resubmission date.

- Include a "Response to the editors and reviewers" document detailing, point-by-point, how you addressed each editor and referee comment. If no action was taken to address a point, you must provide a compelling argument. This response will be used by the editors and reviewers to evaluate your revision.
- Highlight all changes made to your manuscript or provide us with a version that tracks changes.

[REDACTED]

We look forward to seeing the revised manuscript and thank you for the opportunity to review your work. Please do not hesitate to contact me if you have any questions or would like to discuss these revisions further.

Sincerely,

Samantha Antusch

Samantha Antusch, PhD
Editor
Nature Human Behaviour

Reviewer expertise:

Reviewer #1: misinformation/disinformation ; accuracy-increasing interventions and interventions to combat misinformation

Reviewer #3: psychology ; fake news / misinformation

Reviewer #4: political science / political psychology ; partisanship and belief in misinformation / misinformation-sharing ; accuracy-increasing interventions / interventions to combat misinformation

REVIEWER COMMENTS:

Reviewer #1:
Remarks to the Author:

The authors have addressed the issues that I had raised. One more thing that I'd suggest authors should include (after reading Reviewer #2's comment) is "specific" suggestions on interventions that prevent users from falling for misinformation, based on what they found. While the authors acknowledge that it is infeasible to reward users every time they detect misinformation, addressing how their findings build up existing interventions is vital.

Jiyoung Lee

Reviewer #3:
Remarks to the Author:

The authors have adequately addressed my concerns. I believe this manuscript is ready for publication.

Dustin Calvillo

Reviewer #4:
Remarks to the Author:

I thank the authors for their thoughtful review of the paper "Accuracy and Social Motivations Shape Judgements of (Mis)Information". The paper now acknowledges the inconclusive results concerning the hypothesised interaction effects and uses more precise language in reporting results. I also found the new experiment to be a good addition to the paper.

I remain fascinated with the puzzle why this paper arrives to such a different conclusion on the role of partisan motivations than some prior works. One thing I realised upon re-reading the paper is that the present manuscript operationalizes partisan bias as "the number of politically-congruent headlines participants rated as true minus the number of politically-incongruent headlines participants rated as true." (139). Meanwhile, some influential prior work, (e.g. Pennycook & Rand 2019 Cognition) operationalizes it as a difference in truth discernment between congruent and incongruent news. I believe this is different from ideological asymmetries in average truth discernment (across both congruent and incongruent news), which is reported in Figs 3-4 in the present manuscript. I don't have strong views on whether and how this should influence the manuscript. Perhaps a brief justification for the present operationalization and/or an acknowledgement that it is not the only way to think about partisan bias would be easy and beneficial. But of course the authors could also choose to reanalyze their data with the alternative partisan bias measure and see if it's there and if it's affected by the treatments. Then again, perhaps that's beyond the scope of the present paper.

Finally a small thing: it seems to me that the sentence in line 441 contradicts previous statements (and misrepresents a central finding). The effects were only found for *true* headlines, right?

Author Rebuttal, first revision:

REVIEWER COMMENTS:

Reviewer #1:

Remarks to the Author:

The authors have addressed the issues that I had raised. One more thing that I'd suggest authors should include (after reading Reviewer #2's comment) is "specific" suggestions on interventions that prevent users from falling for misinformation, based on what they found. While the authors acknowledge that it is infeasible to reward users every time they detect misinformation, addressing how their findings build up existing interventions is vital.

Jiyoung Lee

Thank you, we are very glad to hear that we have addressed all the changes you have raised. We now mention more specific interventions to prevent users from falling for and sharing misinformation, citing other research as well as findings from our own research. See our new additions on page 14:

“In line with this, experimental studies have found that providing social rewards for sharing high-quality content and punishments for sharing low-quality content⁷⁴ improves the quality of news people report intending to share. Additionally, making people publicly endorse that the news that they share is accurate⁷⁵, or showing people that fellow ingroup members believe content is misleading⁷⁶ also improves people’s sharing intentions. Future work should continue to explore how to incentivize people to engage with more accurate content online by, for example, emphasizing social norms around accuracy or emphasizing the reputational benefits of sharing accurate content (as in Experiment 4).”

Reviewer #3:

Remarks to the Author:

The authors have adequately addressed my concerns. I believe this manuscript is ready for publication.

Dustin Calvillo

Thank you very much, we are glad you are satisfied with the changes and believe the manuscript is ready for publication.

Reviewer #4:

Remarks to the Author:

I thank the authors for their thoughtful review of the paper "Accuracy and Social Motivations Shape Judgements of (Mis)Information". The paper now acknowledges the inconclusive results concerning the hypothesised interaction effects and uses more precise language in reporting results. I also found the new experiment to be a good addition to the paper.

I remain fascinated with the puzzle why this paper arrives to such a different conclusion on the role of partisan motivations than some prior works. One thing I realised upon re-reading the paper is that the

present manuscript operationalizes partisan bias as "the number of politically-congruent headlines participants rated as true minus the number of politically-incongruent headlines participants rated as true." l139. Meanwhile, some influential prior work, (e.g. Pennycook & Rand 2019 Cognition) operationalizes it as a difference in truth discernment between congruent and incongruent news. I believe this is different from ideological asymmetries in average truth discernment (across both congruent and incongruent news), which is reported in Figs 3-4 in the present manuscript. I don't have strong views on whether and how this should influence the manuscript. Perhaps a brief justification for the present operationalization and/or an acknowledgement that it is not the only way to think about partisan bias would be easy and beneficial. But of course the authors could also choose to reanalyze their data with the alternative partisan bias measure and see if it's there and if it's affected by the treatments. Then again, perhaps that's beyond the scope of the present paper.

Thank you for bringing up this interesting point. There has been an ongoing debate in the fake news literature about the term "partisan bias" and the role of partisanship in susceptibility to fake news. We now further justify our measurement of partisan bias. We also note that Pennycook & Rand's measurement of partisan bias that you mention has been called a "problematic conception of partisan bias" and that "partisan bias should be understood in terms of the effect of ideology congruence on overall belief, not truth discernment" (Gawronski, 2021). We note early on that our paper follows these recommendations of how partisan bias should be defined.

For completeness, we nevertheless re-analyzed our data using Pennycook and Rand's measure of partisan bias in terms of truth discernment. Even when we conceptualize partisan bias as the difference in truth discernment between politically-congruent and politically-incongruent news (as Pennycook and Rand do), we still find that our intervention significantly reduces partisan bias. Thus, while we are glad you have encouraged us to discuss this debate over measurements of partisan bias in our revision, this debate does not change any of the main conclusions of our intervention. Although, as you mention, using alternate measures of partisan bias might lead one to downplay the overall effect of partisanship on belief in true or false news, which is ultimately tangential to the main conclusions of our paper.

See our brief mention of this discussion of partisan bias on page 5:

"This measurement of partisan bias follows recommendations from prior work^{15,53}, yet we discuss alternative ways to measure partisan bias and debates about the term "partisan bias"⁵⁴ in

Supplementary Appendix S1. We also re-analyzed our data using an alternate measure of partisan bias in **Supplementary Appendix S1** and found no changes to our main conclusions.”

Also, see our detailed discussion of the debate surrounding the term partisan bias, as well as our re-analysis, on page 6 of the Supplementary Appendix. Note that we chose to include this lengthy section in the supplement due to Nature Human Behavior’s word count:

“Alternate Measures of Partisan Bias. Throughout this article, we follow past research (Batailler et al., 2021; Gawronski, 2021) by defining partisan bias as belief in politically congruent news minus belief in politically-incongruent news. Other work has defined partisan bias as the difference in truth discernment between politically congruent and politically-incongruent news (Pennycook & Rand, 2018, 2021b), which may have led to different conclusions about how much partisanship plays a role in shaping belief in true and false news headlines. However, a recent article argued that measuring partisan bias in terms of truth discernment is a “problematic conception of partisan bias,” as it obscures the effects of partisanship on belief, and that “partisan bias should be understood in terms of the effect of ideology congruence on overall belief, not truth discernment” (Gawronski, 2021). Following the recommendations of this article, we also define partisan bias as the effect of political congruence on belief as opposed to defining it in terms of truth discernment. Using this definition of partisan bias, we found partisan bias to be large in our integrative data analysis ($d = 0.80$ for unincentivized participants, and $d = 0.60$ for incentivized participants), which is similar to the effect sizes found in past work (Batailler et al., 2021), and also found that incentives had a consistent impact on partisan bias across studies.

However, we also re-analyzed our data to examine the difference in truth discernment between politically-congruent and politically-incongruent headlines. Replicating prior work (Pennycook & Rand, 2021b), truth discernment was slightly higher for politically-congruent headlines as compared to politically-incongruent headlines when people were not incentivized to be accurate, $t(1066) = 3.81$, $p < 0.001$, $d = 0.12$. When incentivized to be accurate, there was no difference in truth discernment between politically-congruent and politically-incongruent headlines, $t(1024) = -0.40$, $p = 0.691$, $d = -0.01$. Additionally, incentives significantly decreased the difference in truth discernment between politically-congruent and politically-incongruent headlines, $t(2085.13) = -2.94$, $p = 0.003$, $d = 0.13$. In other words, while people are generally better at distinguishing between truth and falsehoods from their own side as opposed to the other side, incentives led people to become just as good at discerning between truth and falsehood from their side and the other side. Thus, even with this alternate conceptualization of partisan bias, incentives had a significant effect on partisan bias. Altogether, while there may be some disagreement among researchers about how to define terms such as partisan bias, with some arguing that

partisan bias is an improper term as partisan differences in belief do not necessarily reflect “bias” (Pennycook & Rand, 2021a), these disagreements do not change our main findings.”

For more context behind this debate over the term “partisan bias,” we refer to a section from Gawronski (2021, *Trends in Cognitive Sciences*) that informed our thinking about how to measure partisan bias:

“Why do Pennycook and Rand dismiss partisan bias as a major factor when the evidence seems so obvious? The main reason for their dismissal is that they rely on a problematic conceptualization of partisan bias in terms of truth discernment. According to Pennycook and Rand, partisan bias should lead to lower truth discernment for concordant compared with discordant news. Yet, the reviewed evidence suggests the opposite, in that truth discernment is higher, not lower, for concordant compared with discordant news. This finding led the authors to dismiss partisan bias as a major factor in the identification of fake news.

However, a closer analysis reveals that partisan bias should be understood in terms of the effect of ideology congruence on overall belief, not truth discernment. In fact, differential truth discernment for concordant and discordant news is entirely irrelevant for partisan bias, because partisan bias can be present without any effect of ideology congruence on truth discernment (i.e., accuracy in discriminating between real news and fake news). The reason for this is that partisan bias is associated with higher accuracy in judgments of one type of news (real versus fake) and lower accuracy in judgments of the respective other type, leading to an overall null effect on truth discernment (Table 1). For ideology-congruent news, partisan bias is associated with higher accuracy for real news and lower accuracy for fake news. Conversely, for ideology-incongruent news, partisan bias is associated with higher accuracy for fake news and lower accuracy for real news.”

It should be noted that Pennycook & Rand (2021, *Trends in Cognitive Sciences*) have a detailed reply to this commentary. They take issue with the term “partisan bias,” as they do not believe that the partisan divide in belief necessarily represents a bias:

“It may not be justified to refer to this as a ‘bias’, which typically implies that some sort of error has been made. In fact, as noted, our data indicate that people are more accurate when judging political concordant news than politically discordant news, thus, ‘partisan bias’, in this context, is associated with an increase in overall accuracy.”

We now cite these papers and note this debate briefly in our revised paper.

Finally a small thing: it seems to me that the sentence in line 441 contradicts previous statements (and misrepresents a central finding). The effects were only found for *true* headlines, right?

You are correct. The sentence “While effects were only found for false (and not true) headlines, people tend to encounter blatantly false news very infrequently” was a typo. Thank you very much for catching it. We found effects for true news – not false news. We have now removed this sentence, but say earlier in our discussion (on page 14): “No significant effects were found for false news, which people encounter relatively infrequently online⁶⁷.”

Decision Letter, second revision:

Our ref: NATHUMBEHAV-22010105B

1st December 2022

GI

Dear Dr. Rathje,

Thank you for submitting your revised manuscript "Accuracy and Social Motivations Shape Belief in (Mis)Information" (NATHUMBEHAV-22010105B). It has now been seen by the original referees and their comments are below. As you can see, the reviewers find that the paper has improved in revision. We will therefore be happy in principle to publish it in Nature Human Behaviour, pending minor revisions to satisfy the referees' final requests and to comply with our editorial and formatting guidelines.

We are now performing detailed checks on your paper and will send you a checklist detailing our editorial and formatting requirements within a week. Please do not upload the final materials and make any revisions until you receive this additional information from us.

Sincerely,

Samantha Antusch

Samantha Antusch, PhD
Senior Editor
Nature Human Behaviour

Reviewer #4 (Remarks to the Author):

I thank the authors for their thoughtful response to my comments. In my opinion, the revisions have improved the paper, and I am happy to endorse it for publication.

C

Final Decision Letter:

Dear Dr Rathje,

We are pleased to inform you that your Article "Accuracy and Social Motivations Shape Belief in (Mis)Information", has now been accepted for publication in *Nature Human Behaviour*.

Please note that *Nature Human Behaviour* is a Transformative Journal (TJ). Authors whose manuscript was submitted on or after January 1st, 2021, may publish their research with us through the traditional subscription access route or make their paper immediately open access through payment of an article-processing charge (APC). Authors will not be required to make a final decision about access to their article until it has been accepted. IMPORTANT NOTE: Articles submitted before January 1st, 2021, are not eligible for Open Access publication. Find out more about Transformative Journals

With best regards,

Samantha Antusch

Samantha Antusch, PhD
Senior Editor
Nature Human Behaviour